# The lncRNA *Malat1* inhibits miR-15/16 to enhance cytotoxic T cell activation and memory cell formation

Benjamin D Wheeler[1,2], John D Gagnon[1,2], Wandi S Zhu[1,2], Priscila Muñoz-Sandoval[1,2], Simon K Wong[1], Dimitre S Simeonov[1], Zhongmei Li[3], Rachel DeBarge[1,3,4], Matthew H Spitzer[1,3,4,5,6], Alexander Marson[1,3,7], K Mark Ansel[1,2]*

[1]Department of Microbiology & Immunology, University of California San Francisco, San Francisco, United States; [2]Sandler Asthma Basic Research Program, University of California, San Francisco, San Francisco, United States; [3]Gladstone-UCSF Institute of Genomic Immunology, San Francisco, United States; [4]Department of Otolaryngology-Head and Neck Surgery, University of California San Francisco, San Francisco, United States; [5]Parker Institute for Cancer Immunotherapy, San Francisco, San Francisco, United States; [6]Chan Zuckerberg Biohub, San Francisco, United States; [7]Department of Medicine, University of California San Francisco, Lexington, United States

*For correspondence:
mark.ansel@ucsf.edu

**Abstract** Proper activation of cytotoxic T cells via the T cell receptor and the costimulatory receptor CD28 is essential for adaptive immunity against viruses, intracellular bacteria, and cancers. Through biochemical analysis of RNA:protein interactions, we uncovered a non-coding RNA circuit regulating activation and differentiation of cytotoxic T cells composed of the long non-coding RNA *Malat1* (Metastasis Associated Lung Adenocarcinoma Transcript 1) and the microRNA family miR-15/16. miR-15/16 is a widely and highly expressed tumor suppressor miRNA family important for cell proliferation and survival. miR-15/16 play important roles in T cell responses to viral infection, including the regulation of antigen-specific T cell expansion and memory. Comparative Argonaute-2 high-throughput sequencing of crosslinking immunoprecipitation (AHC) combined with gene expression profiling in normal and miR-15/16-deficient mouse T cells revealed a large network of hundreds of direct miR-15/16 target mRNAs, many with functional relevance for T cell activation, survival and memory formation. Among these targets, *Malat1* contained the largest absolute magnitude miR-15/16-dependent AHC peak. This binding site was among the strongest lncRNA:miRNA interactions detected in the T cell transcriptome. We used CRISPR targeting with homology directed repair to generate mice with a 5-nucleotide mutation in the miR-15/16-binding site in *Malat1*. This mutation interrupted *Malat1*:miR-15/16 interaction, and enhanced the repression of other miR-15/16 target genes, including CD28. Interrupting *Malat1* interaction with miR-15/16 decreased cytotoxic T cell activation, including the expression of interleukin 2 (IL-2) and a broader CD28-responsive gene program. Accordingly, *Malat1* mutation diminished memory cell persistence in mice following LCMV Armstrong and *Listeria monocytogenes* infection. This study marks a significant advance in the study of long non-coding RNAs in the immune system by ascribing cell-intrinsic, sequence-specific in vivo function to *Malat1*. These findings have implications for T cell-mediated autoimmune diseases, anti-viral and anti-tumor immunity, as well as lung adenocarcinoma and other malignancies where *Malat1* is overexpressed.

### eLife assessment

This is an **important** study that revealed a new noncoding RNA regulatory circuit involved in T cell function. The authors provide **compelling** evidence, that is more rigorous than the state-of-the-art, using genetically engineered mice and cell-based experiments. The interpretation of the results should be tempered due to the small effect size observed.

## Introduction

Cytotoxic T cells are indispensable for mounting an adaptive immune response against intracellular pathogens and the clearance of mutated cells such as cancer. Over the course of a viral infection, cytotoxic T cells are primed by antigen presenting cells and undergo extensive rounds of intense proliferation (*Murali-Krishna et al., 1998*). As they clonally expand, these cells differentiate into effector and memory cells and acquire effector functions, including the production of cytotoxins and critical cytokines such as IL-2, tumor necrosis factor alpha (TNFα), and interferon gamma (IFNγ) (*Bachmann et al., 1999*). As the infection is cleared, the cytotoxic T cell response contracts as many cells die via apoptosis, yielding a long lived pool of memory cells poised for secondary expansion and protection against reinfection with the same pathogen (*Badovinac et al., 2002*; *Wherry and Ahmed, 2004*). The factors that control this expansion, differentiation, and contraction have been intensely researched in the past decades, focusing in large part on proteins such as transcription factors, signaling enzymes, and cytokines (*Chen et al., 2018*). However, it is of recent interest to understand how non-protein-coding regions of the genome contribute to the regulation of these cells as well. These non-protein-coding elements can be regulatory in nature such as enhancers (*Roychoudhuri et al., 2016*; *Shapiro et al., 1997*), or RNA species that are transcribed but not translated. Two such species of interest to our present study are microRNAs (miRNAs) and long non-coding RNAs (lncRNAs).

MicroRNAs are short (21 nucleotide) RNAs which, when loaded into Argonaute (Ago) proteins, can target mRNAs that contain complementary seed sequences in their 3′ untranslated regions (UTRs) for translation inhibition and degradation (*Bartel, 2018*; *Djuranovic et al., 2012*; *Eichhorn et al., 2014*). In particular, the miR-15/16 family are potent regulators of cell cycle and survival (*Liu et al., 2008*). Previous work from our group utilized conditional deletion of the *Mirc10* and *Mirc30* loci that each contain two of the four major miR-15/16 family members (miR-15a, miR-15b, miR-16-1, and miR-16-2) driven by CD4-cre transgene (hereafter referred to as miR-15/16$^{\Delta/\Delta}$ mice or T cells) to demonstrate that this miRNA family has important effects on cytotoxic T cells (*Gagnon and Ansel, 2019*). In response to viral infection, miR-15/16$^{\Delta/\Delta}$ mice generate more viral-antigen-specific T cells, and these cells preferentially differentiate into memory cells that express CD127 and CD27 (*Gagnon and Ansel, 2019*).

In contrast to the well-defined roles of miRNAs, lncRNAs as a class do not have a single defined function. They are broadly defined as RNAs transcribed by polymerase II, over 200 nucleotides (nt) in length, that lack a translated open reading frame (*Su et al., 2021*; *Wilusz et al., 2009*). Some lncRNAs bind to chromatin and regulate nearby or distant genes, and others scaffold transcription factors and other protein complexes (*Kopp and Mendell, 2018*). Most relevant to the present study, some lncRNAs act as competing endogenous RNAs (ceRNAs) that bind to miRNAs, preventing their binding to and subsequent repression of mRNA targets (*Poliseno et al., 2010*; *Su et al., 2021*). However, rigorous connections between lncRNA physiological functions and molecular mechanisms of action have been hampered by the lack of precise tools to facilitate their study (*Ponting and Haerty, 2022*). As lncRNAs lack an open reading frame, insertion or deletion (indel) mutations do not reliably create null alleles. Instead, investigators have frequently deleted large genomic regions around the promoter or excised the genomic locus of the transcript altogether. These approaches risk disrupting cis-regulatory elements or topologically associated domains that control other nearby genes. Other studies that identified physiological roles for lncRNAs rely on RNAi-mediated lncRNA degradation. Both this approach and genetic manipulations that block lncRNA transcription leave a mechanistic gap between sequence features and downstream function. Studies demonstrating sequence-dependent function of lncRNAs are comparatively rare (*Carrieri et al., 2012*; *Elguindy and Mendell, 2021*; *Faghihi et al., 2008*; *Gong and Maquat, 2011*; *Kleaveland et al., 2018*; *Lee et al., 1999*; *Yoon et al., 2012*).

One lncRNA that has garnered much attention is the Metastasis Associated Lung Adenocarcinoma Transcript 1 (*Malat1*). *Malat1* was first identified as highly expressed in both malignant tumors

and healthy lung and pancreas with high interspecies conservation (*Ji et al., 2003*). *Malat1* is mostly localized to the nucleus, where it is found within nuclear speckles, although it is not necessary for their formation (*Nakagawa et al., 2012*). These characteristics have generated numerous hypotheses about the function of *Malat1* including that it scaffolds epigenetic and splicing factors and acts as a ceRNA to inhibit a variety of miRNAs (*Su et al., 2021*; *Tripathi et al., 2010*). In the immune system, *Malat1* has been studied in dendritic cells, macrophages, T helper cells, T regulatory cells, and cyto-toxic T cells (*Hewitson et al., 2020*; *Kanbar et al., 2022*; *Masoumi et al., 2019*; *Wu et al., 2018*). *Malat1* is actively regulated during cytotoxic T cell differentiation with higher expression in memory precursors and lower expression in short-lived effector cells during LCMV infection (*Kakaradov et al., 2017*). Similarly, T cells stimulated in vitro express decreased levels of *Malat1* over time (*Masoumi et al., 2019*). Early reports using large genetic deletions in the mouse *Malat1* locus detected wide-spread gene expression changes but failed to identify a role for *Malat1* in mouse development or cytotoxic T cell responses (*Nakagawa et al., 2012*; *Yao et al., 2018*; *Zhang et al., 2012*), though a recent report using RNAi detected altered T cell responses (*Kanbar et al., 2022*). Clearly, there remains much to learn about the sequence-specific mechanisms by which *Malat1* tunes pathways and functions essential to cytotoxic T cells.

In the present study, we present a non-coding RNA circuit that regulates the activating signals from CD28 and IL-2, leading to low Bcl-2 expression and loss of memory cells following LCMV infection. We do so by identifying candidate interactions between miRNAs and lncRNAs via a targeted biochem-ical approach, and by creating CRISPR-targeted transgenic mice with precise mutation of *Malat1* to interrogate the physiological function of the interaction between miR-15/16 and *Malat1*. We use this novel mouse to identify CD28-responsive gene programs affected by this circuit, and provide a new sequence-specific function of *Malat1* in vivo.

## Results

### *Malat1* is highly bound by miR-15/16

To identify candidate non-coding ceRNAs in cytotoxic T cells, we performed Argonaute-2 high-throughput sequencing of crosslinking immunoprecipitation (Ago2 HITS-CLIP, AHC). Integrating sequence reads across different classes of transcribed genomic annotations revealed that lncRNAs are bound extensively by Ago2, but the median lncRNA had 7.8 times fewer aligned AHC sequence reads compared to 3′ UTRs where miRNAs canonically bind to mRNAs (*Figure 1A*). This difference likely reflects the comparatively low expression of many lncRNAs, and it highlights the relatively low occupancy of Ago2 on most of these transcripts. Nevertheless, there were individual lncRNAs that stood out as highly bound across the transcript. To prioritize lncRNAs for further investigation, we manually curated the transcripts with the largest number of aligned AHC sequence reads. Among the top 10 most highly bound transcripts, 3 overlapped with protein-coding genes, 2 were on the mitochondrial chromosome, and 3 were repetitive annotations. Strikingly, all of the top 10 transcripts were either annotated as rRNA or contained an rRNA repeat element, except the eighth most bound transcript, *Malat1* (*Table 1*). When the same analysis was repeated with rRNA repeats masked, *Malat1* was the second most highly bound transcript (*Table 2*).

*Malat1* has been proposed to inhibit miRNAs as a ceRNA (*Chen et al., 2017*; *Luan et al., 2016*; *Qiao et al., 2018*; *Wang et al., 2019*; *Wu et al., 2018*; *Xiao et al., 2015*; *Xie et al., 2017*). We used the Piranha peak calling algorithm to identify sites with the highest degree of miRNA binding, as indicated by AHC sequence read number and density (*Figure 1B*). Within cytotoxic T cells, the algo-rithm identified 55 AHC peaks in *Malat1*, the largest of which was extremely pronounced (*Figure 1C*). Compared with other lncRNA-binding peaks, this peak had the 36th most total aligned reads and the 142nd highest read density (15th and 16th, respectively, when rRNA reads are masked). Even when compared to peaks in 3′ UTRs of mRNAs, the largest peak in *Malat1* was the 100th most (98th percen-tile) bound peak in terms of read density (*Figure 1B*).

miRTarget, the custom miRNA-binding prediction algorithm (*Liu and Wang, 2019*), identified an 8-mer seed-binding sequence of the miR-15/16 family centered within the most densely bound region of the called peak (*Figure 1C*). miR-15/16 has not been previously shown to interact with *Malat1* in mice, so to determine whether this peak is miR-15/16 dependent, we performed AHC with cultured CD8$^+$ miR-15/16$^{\Delta/\Delta}$ T cells. In the absence of miR-15/16, Ago2 binding to *Malat1* was

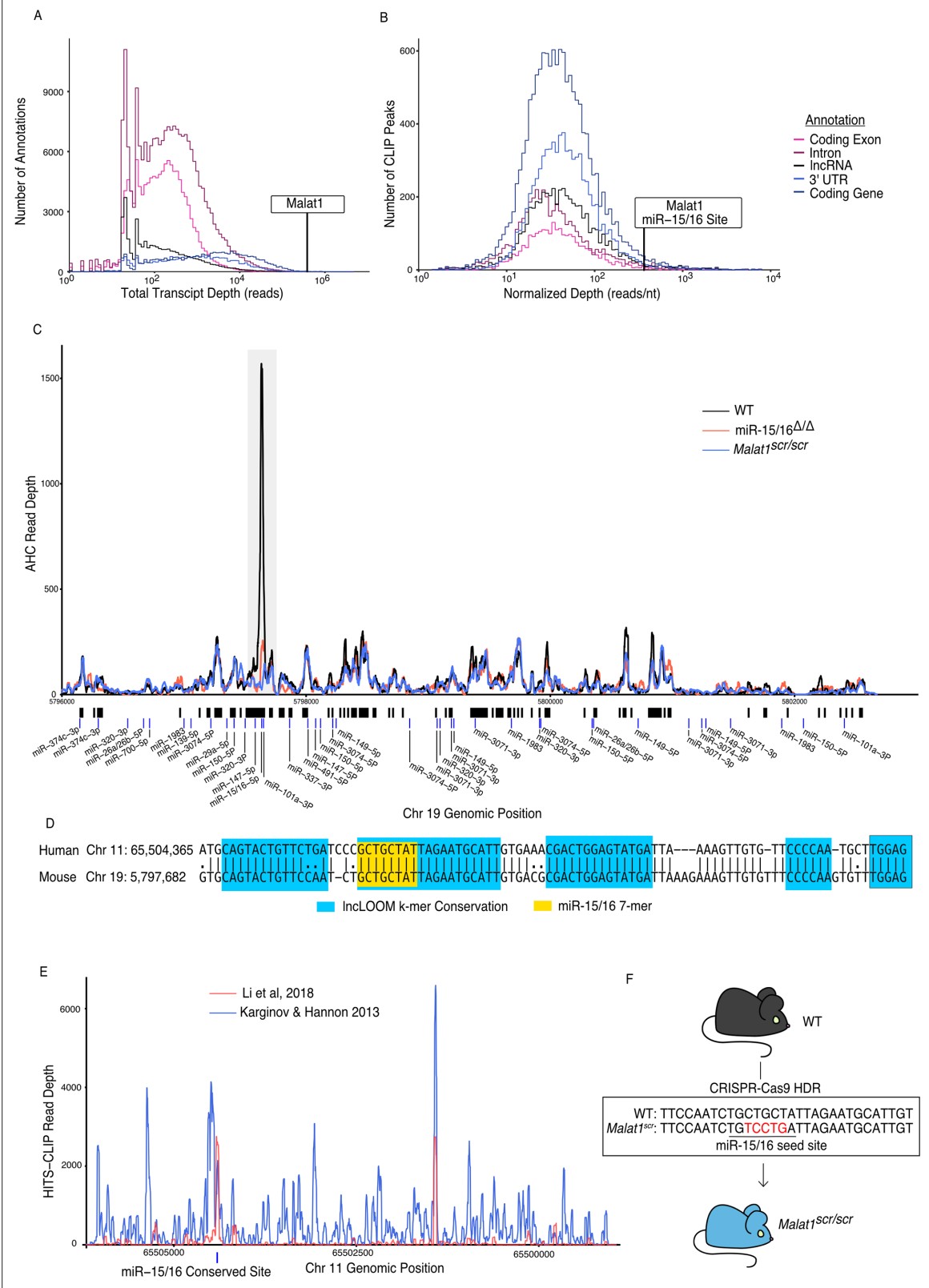

**Figure 1.** *Malat1* is highly bound by miR-15/16. CD8[+] T cells were isolated from spleens, grown in vitro for 5 days, then Ago2 transcriptomic occupancy was assayed via Ago2 HITS-CLIP. (**A, B**) Transcriptome wide analysis of Ago2 HITS-CLIP libraries prepared from WT cells (combined libraries *n* = 2). (**A**) Summed reads across entire annotations. Line indicates total reads across the *Malat1* transcript. *Malat1* was #8 most highly bound lncRNA annotation, which was in the top 0.0091% of all lncRNA annotations analyzed with >0 HITS-CLIP reads. (**B**) Ago2 HITS-CLIP peaks were identified

*Figure 1 continued on next page*

*Figure 1 continued*

and reads were summed within those called peaks that intersected with the given annotation. Peaks were of variable length so summed reads were normalized by peak length. Line indicates HITS-CLIP reads per nucleotide in the called peak containing the miR-15/16-binding site in *Malat1*. This peak was the #121 most bound HITS-CLIP peak in lncRNA peaks analyzed, which was in the top 2.3% of all evaluated peaks in lncRNAs. (**C**) Ago2 HITS-CLIP binding to the mouse *Malat1* locus reads from combined libraries shown (*n* = 2 for each genotype). Gray bar indicates the peak containing the miR-15/16-binding site. Black bars indicate regions identified as peaks by piranha. Blue bars indicate predicted binding sites of miRNAs expressed in our dataset from the miRTarget custom sequence prediction algorithm. Gray bar indicates miR-15/16-binding peak. (**D**) Local alignment of the human and mouse *Malat1* sequences near the miR-15/16 conserved binding site. Highlighting indicates the depth of evolutionary conservation of k-mers as predicted by the lncLOOM algorithm (***Ross et al., 2021***). (**E**) Ago2 HITS-CLIP binding to the human *MALAT1* locus from publicly available datasets (***Karginov and Hannon, 2013***; ***Li et al., 2018***). Blue vertical bar indicates the conserved miR-15/16-binding site. (**F**) Schematic representing the creation of the *Malat1scr* allele. Bases in red indicate the five nucleotides whose sequence was scrambled by CRISPR-Cas9 HDR to prevent miR-15/16 binding.

preserved throughout the whole transcript except for the peak containing the predicted miR-15/16 8-mer seed-binding sequence (***Figure 1C***). We conclude that miRNAs of the miR-15/16 family bind abundantly to *Malat1*. Furthermore, miR-15/16 binding occurred in a region of high evolutionary conservation (***Figure 1D***). The corresponding region containing the miR-15/16-binding site in human *MALAT1* was highly enriched in two publicly available AHC datasets obtained using the 293 human embryonic kidney (HEK) cell line, consistent with Malat1 regulating miR-15/16 in a colorectal cancer cell line (***Karginov and Hannon, 2013***; ***Ji et al., 2019***; ***Li et al., 2018***; ***Figure 1E***). We conclude that miR-15/16 bind abundantly to *Malat1* in mouse and human cells.

We hypothesized that *Malat1* may inhibit the function of miR-15/16 in cytotoxic T cells. To directly address this question in mice, we used CRISPR-Cas9 with homology directed repair to generate mice in which five nucleotides of the miR-15/16 seed-binding sequence within *Malat1* were scrambled (***Figure 1F***). Mice homozygous for this mutation are subsequently referred to as *Malat1scr/scr*. To confirm the targeted functional outcome of this mutation, AHC was performed on cultured CD8[+] T cells isolated from *Malat1scr/scr* mice. Ago2 binding was preserved across the *Malat1* transcript except at the mutated miR-15/16-binding site, where AHC sequence read density was greatly reduced (***Figure 1C***). This finding confirmed that miR-15/16 binds to *Malat1* at this site in a sequence-dependent manner, and provided us with a tool for highly specific investigation of the functional consequences of *Malat1*:miR-15/16 interaction.

### *Malat1* inhibits miR-15/16 availability and activity

*Malat1*:miR-15/16 interaction could lead to regulation and/or degradation of the lncRNA, the miRNAs, or both. To assess whether miR-15/16 degrades *Malat1* we compared the expression of *Malat1* by mRNA sequencing in primary mouse CD8[+] T cells. *Malat1* expression was unchanged in miR-15/16[Δ/Δ]

**Table 1.** lncRNAs with the most AHC reads.

| Non-code gene ID | Total reads | Chromosome | Alias | Note |
|---|---|---|---|---|
| *NONMMUG018330.3* | 4,766,634 | 17 | *ENSMUST00000198477.1* | Contains rRNA repeat |
| *NONMMUG044354.2* | 2,291,785 | M | *mt-Rnr2* | Overlaps *mt-ND1* |
| *NONMMUG044353.2* | 1,208,170 | M | *mt-Rnr1* | |
| *NONMMUG076321.1* | 1,012,732 | 6 | | Overlaps and best aligns to *NONMMUG034479.2* |
| *NONMMUG034478.3* | 1,012,015 | 6 | | Overlaps and best aligns to *NONMMUG034479.2* |
| *NONMMUG034479.2* | 1,011,610 | 6 | | Contains rRNA repeat |
| *NONMMUG007938.3* | 785,970 | 11 | | Intronic of *Gm36876*, contains rRNA repeat (***Weirick et al., 2016***) |
| *NONMMUG020671.2* | 383,447 | 19 | *Malat1* | |
| *NONMMUG015781.2* | 336,664 | 16 | | Intronic of *Zc3h7a*, contains rRNA repeat |
| *NONMMUG044321.2* | 317,822 | 9 | | Contained in *Lars2* 3' UTR, contains rRNA repeat |

**Table 2.** lncRNAs with the most AHC reads that do not align to rRNA repeat elements.

| Non-code gene ID | Total reads | Chromosome | Alias | Note |
|---|---|---|---|---|
| NONMMUG018330.3 | 688,875 | 17 | ENSMUST00000198477.1 | Partially anti-sense to XR_877120.2, binding extends beyond rRNA repeat |
| NONMMUG020671.2 | 383,447 | 19 | Malat1 | |
| NONMMUG094408.1 | 296,723 | 6 | | Contains B4A/B3 SINE, binding restricted to SINE |
| NONMMUG005751.2 | 291,393 | 11 | | Intronic of Dock2, contains multiple RLTR44-int repeat elements, binding restricted to RLTRs |
| NONMMUG094727.1 | 289,326 | 6 | | Contains B4A/B3 SINE, binding is restricted to repeats |
| NONMMUG014644.3 | 241,005 | 15 | Pvt1 | Many repeats, but binding is not restricted to any definite subset |
| NONMMUG096664.1 | 229,834 | 15 | Pvt1 | Splice variant of NONMMUG014644.3 |
| NONMMUG024327.3 | 222,686 | 2 | Oip5os1, Cyrano | Well described in **Han et al., 2020** |
| NONMMUG026716.2 | 207,631 | 3 | | Overlapping Mbnl1 |
| NONMMUG094659.1 | 196,269 | 6 | | Overlapping Foxp1 |

cells as well as in the *Malat1$^{scr/scr}$* cells (**Figure 2A**). Previous studies investigating *Malat1* and other lncRNAs as ceRNA inhibitors of miRNAs have suggested that inhibition occurs by either target RNA-directed miRNA degradation (TDMD) or stoichiometric sequestration of the miRNA from protein-coding mRNA targets (**Han et al., 2020**; **Su et al., 2021**). Therefore, we also tested the possibility that *Malat1* lowers miR-15/16 abundance by TDMD or a related mechanism. However, in freshly isolated mouse CD8$^+$ T cells, miR-15b and miR-16 were unchanged and only miR-15a, a family member with lower expression, was modestly decreased in *Malat1$^{scr/scr}$* cells (**Figure 2B**). We conclude that Malat1 and the miR-15/16 family do not influence each other's absolute abundance in this setting. To test whether *Malat1* affects miR-15/16 function, we examined miR-15/16 target binding in our AHC data. We first defined an experimentally supported list of TargetScan predicted miR-15/16-binding sites with at least one AHC read in both WT and *Malat1$^{scr/scr}$* cells. Using this list, we then examined the read depth at these sites in both *Malat1$^{scr/scr}$* and WT cells (603 sites contained in 479 genes). In WT cells, these sites constituted on average 3.2% of the binding in each 3' UTR, and this figure increased to 3.4% in *Malat1$^{scr/scr}$* cells, indicating that Ago2 occupancy preferentially increased at these sites when miR-15/16 binding to *Malat1* was eliminated (**Figure 2C**). To confirm that binding to these target sites was miR-15/16 dependent, we assessed their AHC read depths in miR-15/16$^{\Delta/\Delta}$ cells as well. As predicted, binding was greatly reduced in miR-15/16$^{\Delta/\Delta}$ cells, representing, on average, 1.0% of binding to a given 3' UTR, a 69% reduction in Ago2 binding compared to WT (**Figure 2D**). In contrast, no significant differences were observed for binding at predicted sites for the highly expressed miRNA families of miR-101, Let-7, miR-21, and miR-142, (**Figure 2—figure supplement 1**). Thus, we conclude that the *Malat1$^{scr}$* allele specifically negatively regulates the first requirement of miR-15/16 function – binding to mRNA targets.

We next sought to investigate whether this increased binding resulted in decreased target mRNA expression. To do so, we analyzed mRNA sequencing data from primary CD8$^+$ T cells 24 hr after stimulation. We generated empirical CDF plots from these data comparing target gene expression in miR-15/16$^{fl/fl}$ to miR-15/16$^{\Delta/\Delta}$ cells as well as WT to *Malat1$^{scr/scr}$* cells. For each of these comparisons, we then compared the distribution for miR-15/16 target genes, as defined above, to the distribution for all other expressed genes. In the case of WT to *Malat1$^{scr/scr}$* cells, the distribution was shifted in favor of decreased target gene expression in the *Malat1$^{scr/scr}$* cells (**Figure 2E**). In the case of miR-15/16$^{fl/fl}$ to miR-15/16$^{\Delta/\Delta}$, the distribution was shifted in favor of increased target gene expression in the miR-15/16$^{\Delta/\Delta}$ cells, indicating that this gene set is relieved of miRNA-induced repression (**Figure 2F**). These data indicate that the increased availability of miR-15/16 leads to increased repression of mRNA target genes when *Malat1*:miR-15/16 interaction is ablated.

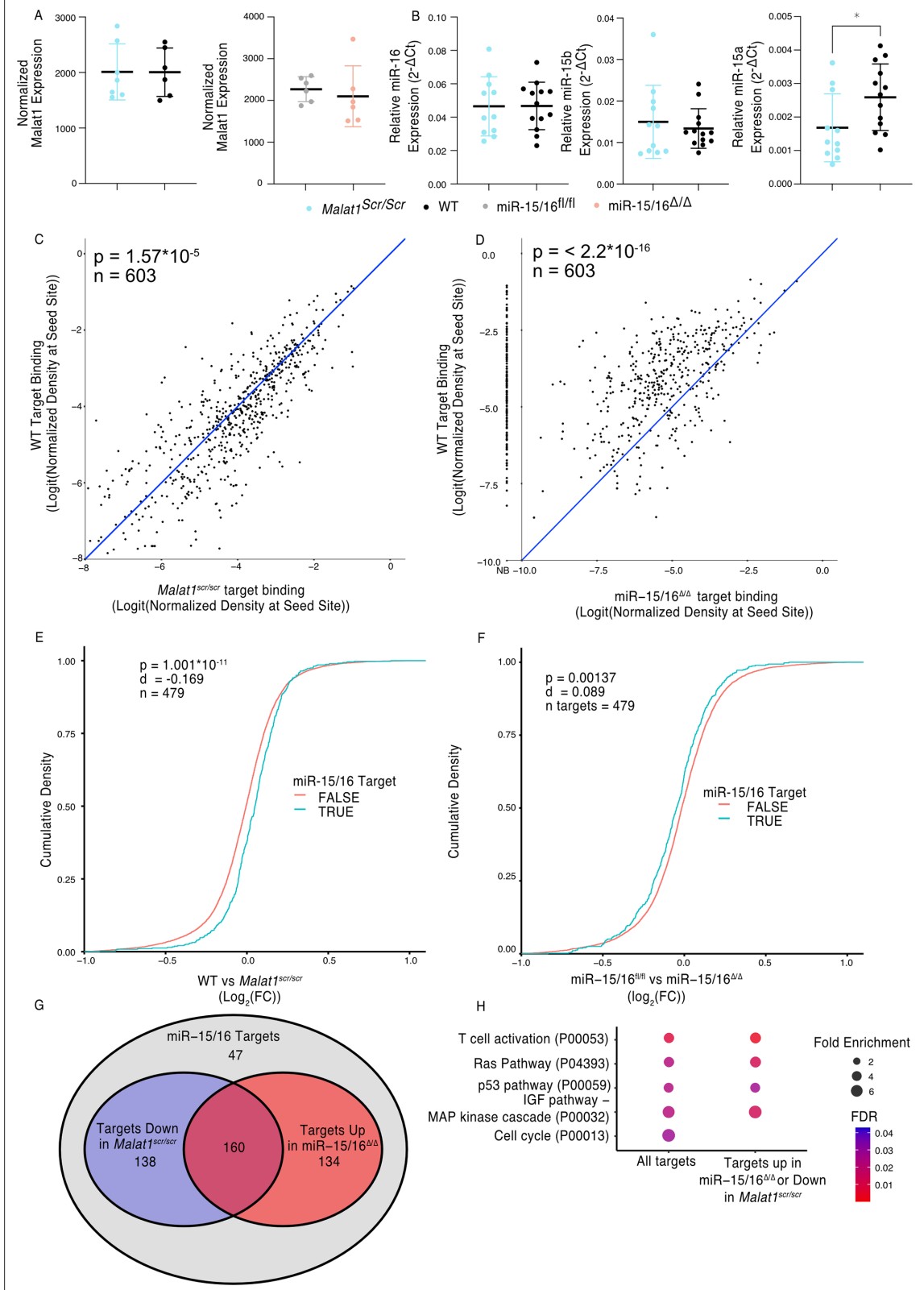

**Figure 2.** *Malat1* Inhibits miR-15/16 binding and suppressive activity. (**A**) *Malat1* expression measure by RNA-seq from CD8⁺ T cells isolated from spleens and stimulated with αCD3 and αCD28 for 24 hr (WT *n* = 6, *Malat1*^scr/scr^ *n* = 7, miR-15/16^fl/fl^ *n* = 6, miR-15/16^Δ/Δ^ *n* = 6; 1 experiment, error bars indicate standard deviation). (**B**) miR-16, miR-15b, and miR-15a expression measured by miRNA qPCR from CD8⁺ T cells freshly isolated from spleens. Expression was determined relative to 5.8 s ribosomal RNA expression. Unpaired *t*-test performed to determine significance, and error bars indicate

*Figure 2 continued*

standard deviation. (**C, D**) TargetScan predicted miR-15/16-binding sites that contained at least one HITS-CLIP read in both WT and *Malat1*$^{scr/scr}$ CD8$^+$ T cells were compared for depth of Ago2 HITS-CLIP reads. First, reads at the predicted seed site were normalized by total Ago2 HITS-CLIP reads in a given 3′ untranslated region (UTR). To best visualize all sites, logit transforms of these values are plotted. Paired *t*-test performed to determine significance. Blue line indicates the identity line. Data for each genotype are from combined libraries of *n* = 2 biological replicates. (**C**) Comparison of WT and *Malat1*$^{scr/scr}$ cells. (**D**) Comparison of WT and miR-15/16$^{Δ/Δ}$. Values to the left of the *y*-axis labeled with NB indicate there was no bind detected at that site in the miR-15/16$^{Δ/Δ}$ cells. (**E, F**) Cumulative density plots to determine changes in expression of miR-15/16 targets. Targets determined by TargetScan predicted miR-15/16 mRNA targets that had at least one 3′ UTR site with reads in both WT and *Malat1*$^{scr/scr}$ CD8$^+$ T cells. Kolmogorov–Smirnov test used to determine significant differences in the distributions of target and non-target genes. (**E**) comparison of the log$_2$(FC) between WT and *Malat1*$^{scr/scr}$ samples stimulated with αCD3 and αCD28 for 24 hr. (**F**) comparison of the log$_2$(FC) between miR-15/16$^{fl/fl}$ and miR-15/16$^{Δ/Δ}$ samples stimulated with αCD3 and αCD28 for 24 hr. (**G**) Venn diagram of miR-15/16 target expression regulated in concordance with the Malat1-miR-15/16 circuit. The blue circle indicates genes with WT vs *Malat1*$^{scr/scr}$ log$_2$(FC) >0 and the red circle indicates genes with miR-15/16$^{fl/fl}$ vs miR-15/16$^{Δ/Δ}$ log$_2$(FC) <0. The purple overlap indicates genes that meet both conditions and the gray indicates genes that do not meet either condition. (**H**) Gene ontology analysis of the bound target set used above as well as genes regulated in accordance with the *Malat1*-miR-15/16 circuit (WT vs *Malat1*$^{scr/scr}$ log$_2$(FC) >0 or miR-15/16$^{fl/fl}$ vs miR-15/16$^{Δ/Δ}$ log$_2$(FC) <0). Enrichment determined within the Panther pathway annotations (*p < 0.05).

The online version of this article includes the following figure supplement(s) for figure 2:

**Figure supplement 1.** *Malat1*$^{scr}$ allele does not disrupt other miRNA families.

**Figure supplement 2.** AHC and gene expression analyses nominate direct miR-15/16 targets involved in growth signaling pathways.

We further investigated genes that displayed reciprocal expression changes in miR-15/16$^{Δ/Δ}$ and *Malat1*$^{scr/scr}$ cells. Of 479 genes in the bound target list, 432 (90%) were either downregulated in *Malat1*$^{scr/scr}$ cells or upregulated in miR-15/16$^{Δ/Δ}$ cells, compared to controls. Among these genes, the expression of 298 (62%) were decreased in *Malat1*$^{scr/scr}$ cells, and 294 (61%) were increased in miR-15/16$^{Δ/Δ}$ cells, with 160 (33%) both increased in miR-15/16$^{Δ/Δ}$ and decreased in *Malat1*$^{scr/scr}$ cells (***Figure 2G***). Gene ontology enrichment analysis revealed multiple pathways associated with growth factor and antigen receptor signaling affected by the Malat1:miR-15/16 circuit in cytotoxic T cells (***Figure 2H***). Many of these modules, including the T cell activation module, were identified because they share key signaling proteins. For example, gene expression and AHC data support direct miR-15/16 targeting of the costimulatory receptor *Cd28*, the alpha subunit of phosphatidylinositol 3-kinase (*Pik3r1*) and c-Jun N-terminal kinase (JNK1), encoded by *Mapk8* (***Figure 2—figure supplement 2*** and ***Figure 3***). Targeting of these genes by miR-15/16 is consistent with their known tumor suppressor role (***Cimmino et al., 2005***; ***Gagnon and Ansel, 2019***) and with *Malat1*'s association with cancer cell proliferation and metastasis (***Ji et al., 2003***), providing new mechanistic insight into those observations.

## *Malat1* enhances CD28 expression and downstream CD8 T cell activation

Given that the *Malat1*:miR-15/16 circuit regulated genes essential to T cell activation, and in particular CD28, we next investigated the functional consequences of perturbing this circuit in cytotoxic T cells. First, we looked directly at CD28. AHC in WT cells detected a prominent peak in the CD28 3′ UTR at the TargetScan predicted binding site for miR-15/16. This peak was absent in miR-15/16$^{Δ/Δ}$ cells, whereas other binding peaks were preserved, empirically verifying that this binding event is miR-15/16 dependent. AHC in *Malat1*$^{scr/scr}$ cells indicated a modest increase in Ago2 binding at this site compared to WT cells relative to Ago2 binding in the whole 3′ UTR (***Figure 3A***). These data, along with our previous demonstration that the CD28 3′ UTR is miR-15/16 responsive, indicate that CD28 is part of a module of miR-15/16 target genes that are highly likely to be affected by the *Malat1*$^{scr}$ allele (***Gagnon and Ansel, 2019***).

Indeed, CD28 expression and T cell activation were regulated by the *Malat1*:miR-15/16 circuit. Flow cytometric measurement in primary splenic CD8$^+$ T cells revealed that CD28 protein expression was decreased in *Malat1*$^{scr/scr}$ mice and enhanced in miR-15/16$^{Δ/Δ}$ mice (***Figure 3B, C***). To investigate activation-induced gene expression, we performed RNA-seq on splenic CD8$^+$ T cells stimulated for 24 hr with plate-bound αCD3 with or without αCD28 crosslinking antibodies diagrammed in (***Figure 3D***). CD28 stimulation enhances distinct activation-induced gene expression changes in T cells (***Martínez-Llordella et al., 2013***), and these CD28-responsive genes were altered in miR-15/16$^{Δ/Δ}$ and *Malat1*$^{scr/scr}$ cells (***Figure 3E–H***). The previously defined set of 164 genes that are upregulated in WT cells stimulated with αCD3 + αCD28 compared to αCD3 alone was also significantly upregulated

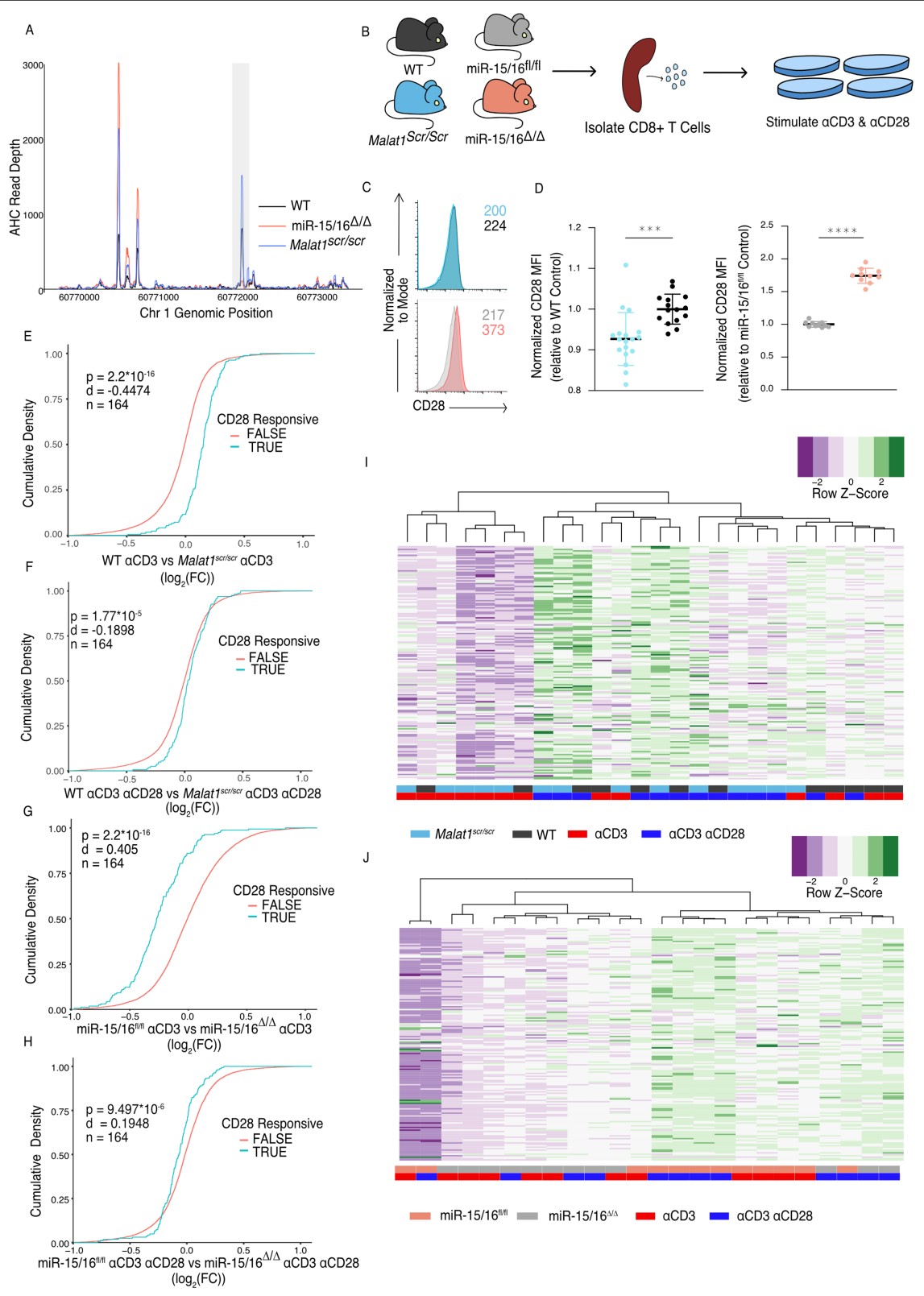

**Figure 3.** The *Malat1*-miR-15/16 circuit increase CD28 expression and costimulation induced gene expression. (**A**) Ago2 HITS-CLIP binding at the Cd28 locus. Sequencing libraries generated from CD8+ T cells isolated from spleens and cultured for 5 days (combined libraries from *n* = 2 for each genotype). Gray bar indicates the peak containing the TargetScan predicted miR-15/16-binding site. (**B**) Schematic illustrating the assay scheme to assay acute gene expression downstream of CD28 costimulation. (**C**) Representative flow cytometry plots of CD28 expression on naive (CD62L+ CD44−)

*Figure 3 continued on next page*

*Figure 3 continued*

CD8[+] T cells from spleens of unchallenged mice. Mean fluorescence intensity for the sample reported in the upper right of the plot. (**D**) Quantification of CD28 mean fluorescence intensity normalized to the relevant control (*Malat1scr/scr* compared to WT from three independent experiments; miR-15/16fl/fl compared to miR-15/16Δ/Δ from two independent experiments). (**E–H**) Cumulative density plots comparing expression of CD28-responsive gene set defined as genes from *Martínez-Llordella et al., 2013* with αCD3αCD28 vs αCD3 log$_2$(FC) >1.5 and adjusted p-value <0.001. Kolmogorov–Smirnov test used to determine significant differences in the distributions of target and non-target genes. αCD3 used at 1 μg/ml, and αCD28 used at 1 μg/ml. (**E**) Comparison of CD28-responsive genes in WT vs *Malat1scr/scr* cells stimulated with αCD3 alone. (**F**) Comparison of CD28-responsive genes in WT vs *Malat1scr/scr* cells stimulated with αCD3 and αCD28. (**G**) Comparison of CD28-responsive genes in miR-15/16fl/fl vs miR-15/16Δ/Δ cells stimulated with αCD3 alone. (**H**) Comparison of CD28-responsive genes in miR-15/16fl/fl vs miR-15/16Δ/Δ cells stimulated with αCD3 and αCD28. (**I, J**) Heatmaps of CD28-responsive gene set expression by genotype and stimulation condition. Dendrograms represent unbiased hierarchical clustering of the samples. (**I**) *Malat1scr/scr* and WT samples compared with αCD3 ± αCD28. (**J**) miR-15/16fl/fl vs miR-15/16Δ/Δ samples compared with αCD3 ± αCD28 ( ***p < 0.001; ****p < 0.0001, error bars indicate standard deviation).

The online version of this article includes the following figure supplement(s) for figure 3:

**Figure supplement 1.** CD28-responsive genes are induced by αCD28 stimulation in all genotypes tested.

in the αCD3 + αCD28 condition compared to αCD3 alone for each genotype tested in our experiments (*Figure 3—figure supplement 1*). Importantly, the Malat1:miR-15/16 circuit affected this gene set in the αCD3 alone condition, with *Malat1scr/scr* cells exhibiting decreased expression compared to WT cells, and miR-15/16Δ/Δ cells exhibiting increased expression compared to miR-15/16fl/fl cells (*Figure 3E, G*). This trend was preserved, but to a lesser degree, in the αCD3 and αCD28 conditions (*Figure 3F, H*). Thus, *Malat1*:miR-15/16 interaction enhanced expression of a costimulation-responsive gene expression program in activated T cells, and it further enhanced expression of that module even when CD28 costimulation was directly engaged.

Unsupervised hierarchical clustering of these samples based on the expression of the 164 costimulation-responsive gene set further underscored the costimulatory-like effect of the Malat1:miR-15/16 circuit. In comparing gene expression in *Malat1scr/scr* and WT samples, three major groups emerged. The group with the lowest average costimulation-responsive gene expression contained only samples stimulated with αCD3 alone and primarily *Malat1scr/scr* samples. The group with intermediate expression was the largest group with an even representation of *Malat1scr/scr* samples and WT samples. While samples in this intermediate group were from both stimulation conditions, the *Malat1sc/scr* samples tended to be from the αCD3 + αCD28 condition and the WT samples had an even representation from both stimulation conditions. The group with the highest average expression contained predominantly αCD3 + αCD28 stimulated samples with an even representation of *Malat1scr/scr* and WT samples (*Figure 3I*). Thus, *Malat1*:miR-15/16 interaction and engagement of CD28 signaling additively induced costimulation-responsive genes. This observation is further supported when clustering the miR-15/16Δ/Δ and miR-15/16fl/fl samples. The two largest clusters were divided nearly exclusively by genotype. The group with lower average expression of costimulation-responsive genes contained predominantly mir-15/16fl/fl samples stimulated with αCD3 alone. The group with the higher average expression contained a majority of miR-15/16Δ/Δ samples, and the only miR-15/16fl/fl samples within the group received αCD3 + αCD28 stimulation. Within the higher expression group, the miR-15/16fl/fl samples stimulated with αCD3 + αCD28 subclustered with miR-15/16Δ/Δ samples stimulated with αCD3 alone (*Figure 3J*). A third, much smaller cluster was composed of both samples from a single outlier biological replicate. Overall, these data show that miR-15/16 restrict costimulation-responsive gene expression, and that *Malat1*:miR-15/16 interaction limits this effect.

In addition to the proximal changes in gene expression, downstream functional outcomes of CD28 costimulation were also affected. Early activation genes have been well described in T cells, with CD69 responding to many cues including TCR and CD28 ligation, and Nur77 responding very specifically to TCR signals (*Ashouri and Weiss, 2017*; *Vandenberghe et al., 1993*). In similar fashion as above, we assessed the expression of these proteins 2 and 4 hr after stimulation with plate-bound cross-linking antibodies on primary CD8[+] T cells. CD69 exhibited decreased expression in *Malat1scr/scr* cells at both 2 and 4 hr post stimulation (2 hr: p = 0.0014 and 4 hr: p = 0.0598) and increased expression in miR-15/16Δ/Δ cells at 4 hr post stimulation only (p < 0.0001) (*Figure 4A, B*), in accordance with costimulation-responsive gene module expression (*Figure 3*). There was no difference, however, in expression of Nur77 across *Malat1scr/scr*, WT, and miR-15/16Δ/Δ cells, indicating that TCR signals were equivalent (*Figure 4A, C*). Another key consequence of CD28 ligation is the production of IL-2 (*Fraser*

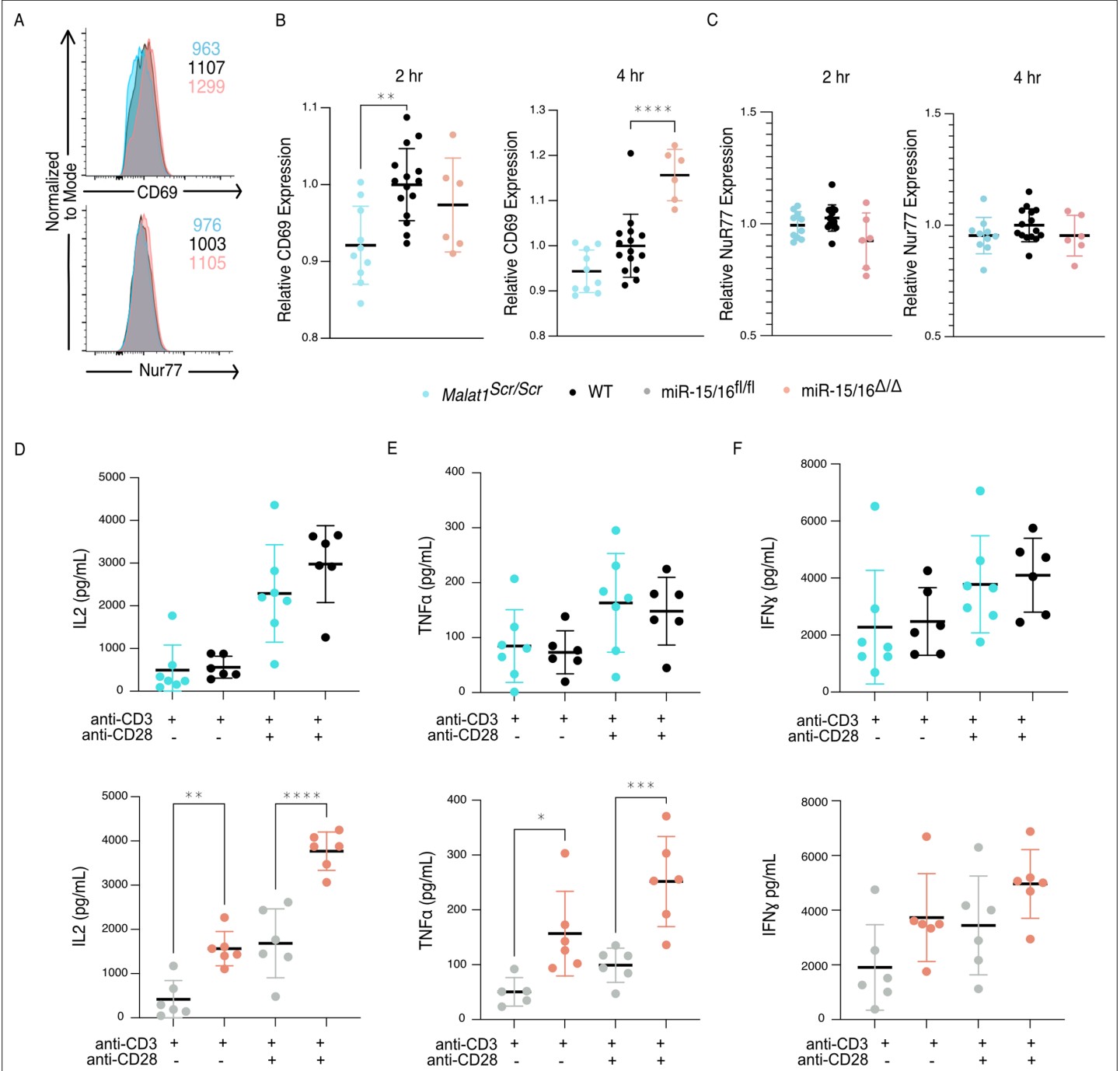

**Figure 4.** The *Malat1*-MiR-15/16 circuit increases functional outcomes of CD28 costimulation. (**A–C**) CD8+ T cells were isolated from spleens and stimulated with αCD3 and αCD28 antibodies, results displayed are gated on activated cells (CD69+ Nur77+). (**A**) Representative histograms of CD69 and Nur77 expression 4 hr after stimulation. (**B**) Quantification of CD69 mean fluorescence intensity 2 and 4 hr after stimulation. Both time points reflect statistically significant changes (p < 0.01) by ordinary one-way analysis of variance (ANOVA); statistics displayed on graph represent results of post hoc multiple comparisons of *Malat1*$^{scr/scr}$ to WT and miR-15/16$^{Δ/Δ}$ to WT. Data from two independent experiments, each normalized to WT average value. (**C**) Quantification of Nur77 mean fluorescence intensity 2 and 4 hr after stimulation. No significant changes determined by ordinary one-way ANOVA. Data from two independent experiments, each normalized to WT average value. (**D–F**) Quantification of cytokine secretion into the supernatant by CD8+ T cells isolated from spleens, stimulated αCD3 ± αCD28, and cultured 16 hr. Cell-free supernatant protein concentration measured by ELISA. Data from a single experiment. (**D**) Quantification of IL-2 secretion. By two-way ANOVA, in both experiments there was a significant (p < 0.0001) increase in IL-2 with the addition of αCD28 stimulation. But the only significant (p = 0.0001) genotypic effect was increased IL-2 secretion in miR-15/16$^{Δ/Δ}$ vs mir-15/16$^{fl/fl}$. Comparisons shown on plot are the results of post hoc multiple comparison tests. (**E**) Quantification of TNFα secretion. By two-way ANOVA, in both

*Figure 4 continued on next page*

**Figure 4 continued**

experiments there was a significant (p < 0.0001) increase in IL-2 with the addition of αCD28 stimulation. But the only significant (p = 0.003) genotypic effect was increased IL-2 secretion in miR-15/16$^{\Delta/\Delta}$ vs mir-15/16$^{fl/fl}$. Comparisons shown on plot are the results of post hoc multiple comparison tests. (**F**) Quantification of IFNγ secretion. By two-way ANOVA, in both experiments there was a significant (p < 0.0001) increase in IL-2 with the addition of αCD28 stimulation. But no genotypic effect was observed (*p < 0.05; **p < 0.01; ***p <0 .001; ****p < 0.0001, error bars indicate standard deviation).

et al., 1991; *Maggirwar et al., 1997*; *Vandenberghe et al., 1993*). Therefore, we assessed cytokine production in the supernatants of these cultures 16 hr after stimulation. In line with previous findings in LCMV-infected mice (*Gagnon and Ansel, 2019*), miR-15/16$^{\Delta/\Delta}$ cells made significantly more IL-2 and TNFα after αCD3 stimulation both with and without αCD28, with a trend toward increased IFNγ as well (*Figure 4D–F*). *Malat1$^{scr/scr}$* cells stimulated with αCD3 + αCD28 exhibited a reciprocal trend specifically for IL-2 (23% decrease compared to WT, p = 0.26) (*Figure 4D*). Together, these data show that the *Malat1*:miR-15/16 circuit regulates key functional outcomes of CD28-mediated costimulation, from proximal gene expression changes to early activation protein expression and cytokine secretion.

### *Malat1* enhances cytotoxic memory T cell differentiation

miR-15/16 restrict memory T cell differentiation, cell cycle, and cell survival during the response to LCMV Armstrong infection (*Gagnon and Ansel, 2019*). CD28 costimulation is essential for IL-2 production, memory cell formation, and memory recall responses in vivo (*Borowski et al., 2007*; *Eberlein et al., 2012*; *Fuse et al., 2008*; *Grujic et al., 2010*; *Suresh et al., 2001*). IL-2 is also essential for these same processes and in particular is required in a CD8$^+$ T cell-intrinsic nature for the formation of CD127$^+$ KLRG1$^-$ memory cells (*Blattman et al., 2003*; *Kahan et al., 2022*; *Pipkin et al., 2010*; *Toumi et al., 2022*; *Whyte et al., 2022*). Since *Malat1* inhibits miR-15/16 activity and this circuit impacts proper T cell activation and IL-2 production after CD28 costimulation, we hypothesized that *Malat1$^{scr/scr}$* cells would exhibit poor memory formation and survival.

We first examined steady-state memory populations of polyclonal T cells in unchallenged, young mice. *Malat1$^{scr/scr}$* mice had normal naive and central memory T cell populations, but a reduced percentage and number of effector memory (CD44$^+$CD62L$^-$) cells in the spleen (*Figure 5—figure supplement 1*). Therefore, we investigated the cell-intrinsic nature of the *Malat1$^{scr/scr}$* memory cell impairment during a viral challenge known to induce a large memory response. We bred the *Malat1$^{scr/scr}$* mice with P14 TCR transgenic mice that express an antigen receptor specific for the immunodominant LCMV GP33 peptide. We then transferred *Malat1$^{scr/scr}$* and WT P14 T cells into congenic CD45.1 hosts followed by LCMV Armstrong infection and tracked the acute and memory responses in the spleen and liver, the sites of primary LCMV infection (*Figure 5A*; *Matloubian et al., 1993*).

The total numbers of P14 cells were similar in recipients of transferred WT or *Malat1$^{scr/scr}$* P14 cells in both the spleen and liver at day 7, and in the spleen at day 31. However, there was a reduction in *Malat1$^{scr/scr}$* P14 cells in the liver at day 31 (*Figure 5B, C*). Despite having no effect on cell expansion at the peak of infection and mixed cell number results in the memory phase, the *Malat1$^{scr}$* allele had a distinct effect on the cellular phenotype across organs and time points. *Malat1$^{scr/scr}$* P14 cells preferentially displayed a phenotype associated with terminally differentiated effector memory cells (t-TEM), defined by KLRG1 expression and lack of CD127 expression as shown by *Milner et al., 2020*, and a corresponding reduction in the percentage of KLRG1$^-$ CD127$^+$ memory cells (*Figure 5D–F*). Previous work has shown that substantial heterogeneity exists in the memory pool and that CD27 and CD43 can be useful in delineating functional differences such as recall potential between different memory cells. For instance, CD27$^+$ memory cells tend to produce more IL-2 than CD27$^-$ cells (*Milner et al., 2020*), and CD43$^+$ CD27$^+$ cells are more effective at clearing *Listeria* upon re-challenge (*Hikono et al., 2007*; *Olson et al., 2013*). We therefore assessed these markers on the transferred *Malat1$^{scr/scr}$* and WT P14 cells. Consistent with increased KLRG1$^+$ CD127$^-$ t-TEM cells there were proportionally more CD27$^-$ CD43$^-$ cells in both the spleen and liver at day 31 among the *Malat1$^{scr/scr}$* transferred cells compared to WT (*Figure 5G, I*). CD27$^+$ CD43$^-$ cells were unchanged. The proportional reduction in CD127$^+$ KLRG1$^-$ cells corresponded with a decreased proportion of CD43$^+$ CD27$^+$ cells. Notably, the total numbers of all CD27$^-$ subsets were unchanged. The proportional differences we observed were primarily driven by a reduction in the number of CD43$^+$ CD27$^+$ *Malat1$^{scr/scr}$* P14 cells compared with wild-type P14 cells (45% reduction in liver (p = 0.0058), 30% reduction in spleen (p = 0.113)) (*Figure 5G–I*). This phenotype is exactly reciprocal to the increase in CD27$^+$ memory cells

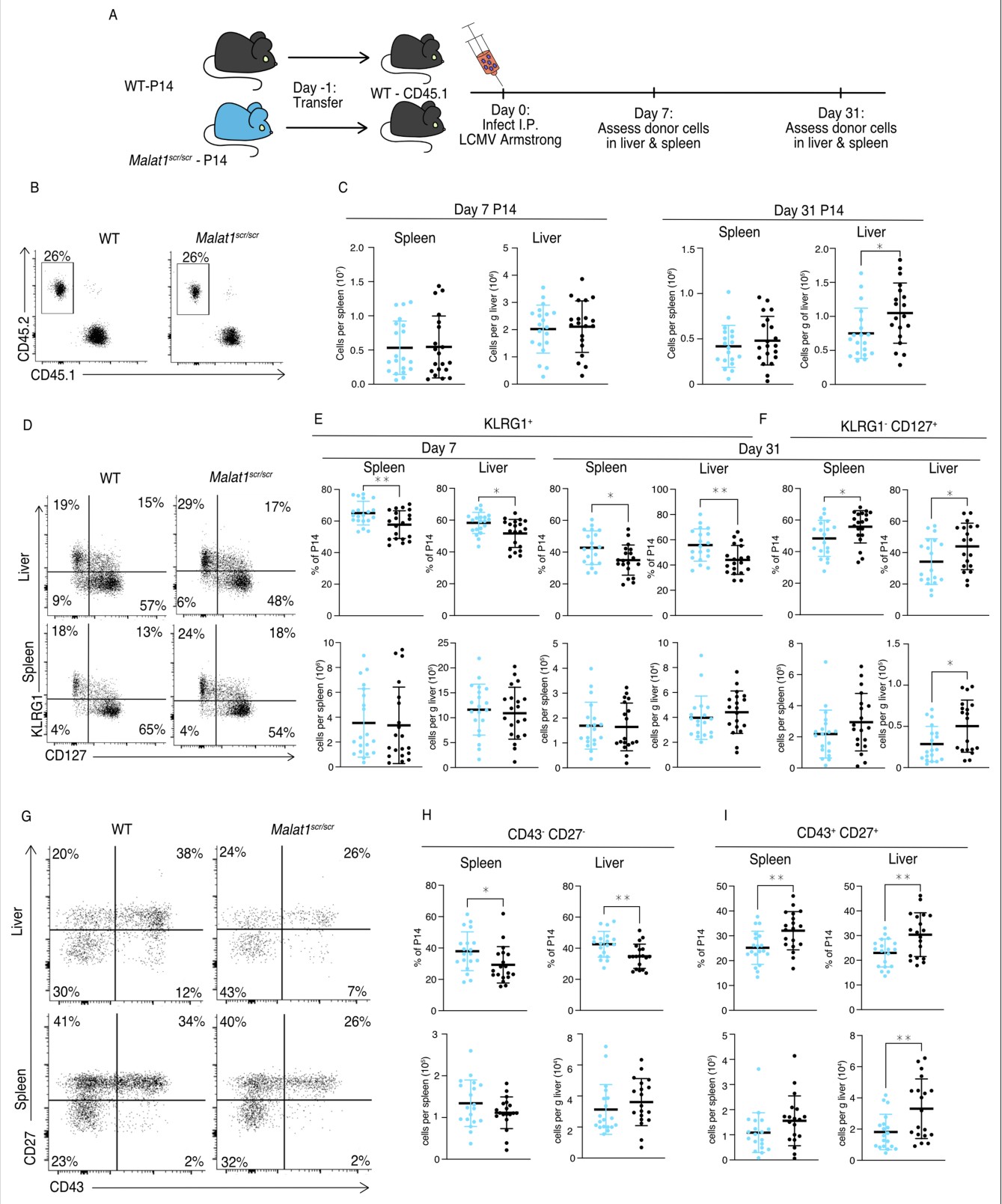

**Figure 5.** *Malat1* enhances memory T cell persistence following LCMV infection. *Malat1^scr/scr* and WT cells containing the GP33-specific TCR transgene (P14) on the CD45.2 background were transferred separately into congenic CD45.1 WT hosts. One day later the recipient mice were infected with 5 × 10^5 p.f.u. I.p. LCMV Armstrong. LCMV-specific responses were assayed by monitoring the transferred cells by flow cytometry in the blood, spleen, and liver over time (data from two independent experiments per time point) (**A**) Schematic of experimental design. (**B**) Representative flow plots to identify

*Figure 5 continued on next page*

*Figure 5 continued*

and quantify transferred cells. (**C**) Quantification of transferred P14 cell numbers at days 7 and 31. (**D**) Representative flow plots of KLRG1 and CD127 expression on P14 cells at day 31 post infection. (**E**) Quantification of P14 KLRG1$^+$ cells by percent of P14 and total numbers in spleen and liver at days 7 and 31 post infection. (**F**) Quantification of P14 KLRG1$^-$ CD127$^+$ by percent of P14 and total numbers in spleen and liver at day 31 post infection. (**G**) Representative flow plots of CD43 and CD27 expression on P14 cells at day 31 post infection. (**H**) Quantification of P14 CD43$^-$ CD27$^-$ t-Tem cells by percent of P14 and total numbers in spleen and liver at day 31 post infection. (**I**) Quantification of P14 CD43$^+$ CD27$^+$ memory cells by percent of P14 and total numbers in spleen and liver at day 31 post infection. Statistics displayed determined by unpaired *t*-test between *Malat1$^{scr/scr}$* and WT transferred cells (*p < 0.05; **p < 0.01, error bars indicate standard deviation).

The online version of this article includes the following figure supplement(s) for figure 5:

**Figure supplement 1.** *Malat1* regulates memory formation in unchallenged poly-clonal animals.

**Figure supplement 2.** *Malat1* is epistatic to miR-15/16 in the regulation memory cell expansion following LCMV infection.

previously documented in miR-15/16$^{\Delta/\Delta}$ mice (*Gagnon and Ansel, 2019*). To test whether the effect of the *Malat1$^{scr}$* allele was epistatic to miR-15/16, we bred *Malat1$^{scr/scr}$* to miR-15/16$^{\Delta/\Delta}$ mice to generate triple mutant *Malat1$^{scr/scr}$* miR-15/16$^{\Delta/\Delta}$ mice. The poly-clonal LCMV response in these mice phenocopied that of miR-15/16$^{\Delta/\Delta}$ with a WT allele of *Malat1*, indicating that the observed effects of the *Malat1$^{scr}$* allele are epistatic to miR-15/16 (*Figure 5—figure supplement 2*). We conclude that Malat1 inhibits miR-15/16 in responding CD8$^+$ T cells during LCMV infection, leading to fewer CD43$^+$ CD27$^+$ memory cells. Compared to LCMV, *Listeria monocytogenes* (LM) infection induces lower expression of multiple costimulatory ligands so the antigen-specific response is more sensitive to the ablation of CD28 costimulation (*Welten et al., 2015*). Therefore, we hypothesized that cytotoxic T cell response may be more impaired by the *Malat1$^{scr}$* allele during LM infection. We first sought to understand the effect of miR-15/16 on cytotoxic T cells during primary LM infection. To do so, we directly infected polyclonal mir-15/16$^{fl/fl}$ and miR-15/16$^{\Delta/\Delta}$ mice with LM expressing the LCMV GP33 peptide (LM-GP33) (*Figure 6A*). In contrast to the increased antigen-specific CD8$^+$ T cell numbers in LCMV-infected miR-15/16$^{\Delta/\Delta}$ mice (*Gagnon and Ansel, 2019*), we observed similar cell numbers in LM-GP33-infected miR-15/16$^{\Delta/\Delta}$ and miR-15/16$^{fl/fl}$ mice (*Figure 6B*). However, the effect on memory cell differentiation was still present with decreased percentages of CD127$^-$ KLRG1$^+$ t-TEMs and increased percentages of CD127$^+$ KLRG1$^-$ memory cells (*Figure 6C, D*). Again, miR-15/16$^{\Delta/\Delta}$ antigen-specific cells had a significantly lower proportion of CD43$^-$ CD27$^-$ cells and a trend toward proportionally more CD43$^+$ CD27$^+$ cells (p = 0.127) (*Figure 6E, F*).

Given that antigen-specific miR-15/16$^{\Delta/\Delta}$ T cells exhibited enhanced memory cell differentiation during LM-GP33 infection, we further tested the role of *Malat1*:miR-15/16 interaction in this model using the P14 adoptive transfer system (*Figure 6F*). Transferred *Malat1$^{scr/scr}$* and WT P14 cells accumulated in the spleen and the liver to similar numbers at both days 7 and 31 following LM-GP33 infection (*Figure 6H*). In contrast to LCMV, there was no change in the proportion or number of CD127$^-$ KLRG1$^+$ *Malat1$^{scr/scr}$* P14 cells at day 7 in the spleen or liver (*Figure 6I*). However, in LM-GP33 infection at day 7, CD43 and CD27 expression did not explicitly mirror the phenotype observed by CD127 and KLRG1 expression. In the spleen, there was a proportional and numerical increase in CD43$^-$ CD27$^-$ P14 cells with the *Malat1$^{scr/scr}$* genotype (*Figure 6J*). There was a decreased proportion of CD43$^+$ CD27$^+$ cells (p = 0.0531), but this was despite a trend toward increased numbers of these cells (p = 0.1056) in the *Malat1$^{scr/scr}$* P14 cells (*Figure 6K*). In the liver, there were no significant trends in any of these populations at day 7 (*Figure 6J, K*).

Although some of the phenotypes observed with LCMV were attenuated or absent during acute infection with LM-GP33 at day 7, Malat1:miR-15/16 interaction had more pronounced effects on memory cell populations at later times post infection (*Figure 6L, M*). On day 31, KLRG1$^+$ cells were increased in proportion and numbers in the *Malat1$^{scr/scr}$* P14 cells (*Figure 6N*). In the liver, a similar trend in KLRG1$^+$ cell proportion existed (p = 0.233), but the numeric effect was entirely absent (*Figure 6N*). Surprisingly, this effect was restricted to the KLRG1$^+$ populations, as CD127$^+$ KLRG1$^-$ memory cells were unaffected in proportion and number in both the spleen and liver (*Figure 6O*). In both organs CD43$^-$ CD27$^-$ t-TEMs were proportionally increased and CD43$^+$ CD27$^+$ memory cells were proportionally decreased in *Malat1$^{scr/scr}$* P14 cells (*Figure 6P, Q*). The numeric underpinnings of these proportional changes were different in each organ, with a 94% increase in CD43$^-$ CD27$^-$ cells in the spleen, and a 40% loss of the CD43$^+$ CD27$^+$ population in the liver (*Figure 6P, Q*). Taken together,

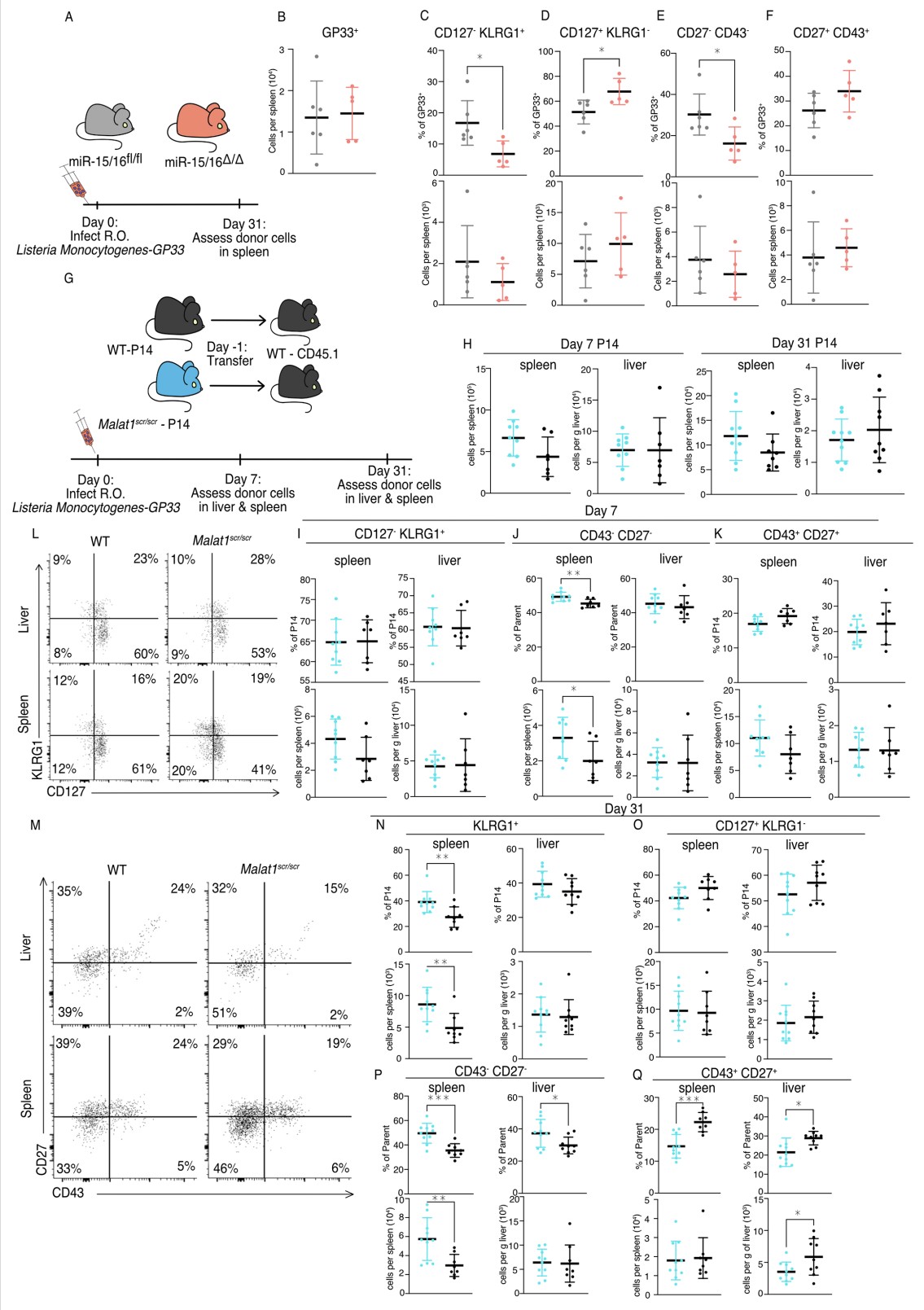

**Figure 6.** *Malat1* and miR-15/16 alter memory T cell differentiation following *Listeria monocytogenes* infection. miR-15/16$^{fl/fl}$ and miR-15/16$^{\Delta/\Delta}$ with a polyclonal TCR repertoire were directly infected with 2 × 10$^4$ colony forming units (c.f.u.) r.o. *Listeria monocytogenes-gp33* (LM-GP33). LM-GP33-specific responses were then assayed in the spleen 31 days post infection (miR-15/16$^{\Delta/\Delta}$ n = 5 and miR-15/16$^{fl/fl}$ n = 6 from a single experiment). (**A**) Schematic of experimental design. (**B**) Quantification of tetramer-specific CD8 T cells in the spleen. Quantification of tetramer-specific subpopulations by percent

*Figure 6 continued on next page*

*Figure 6 continued*

of GP33$^+$ and numbers for (**C**) CD127$^-$ KLRG1$^-$, (**D**) CD127$^+$ KLRG1$^-$, (**E**) CD27$^-$ CD43$^-$, and (**F**) CD27$^+$ CD43$^+$ populations. *Malat1$^{scr/scr}$* and WT cells containing the GP33-specific TCR transgene (P14) on the CD45.2 background were transferred separately into congenic CD45.1 WT hosts. One day later the recipient mice were infected with $2 \times 10^4$ c.f.u. r.o. LM-GP33. LM-GP33-specific responses were assayed by monitoring the transferred cells by flow cytometry in the spleen and liver at discrete time points (data from a single experiment per time point). (**G**) Schematic of experimental design. (**H**) Quantification of transferred P14 cell numbers at days 7 and 31. Quantification of P14 (**I**) CD127$^-$ KLRG1$^+$, (**J**) CD43$^-$ CD27$^-$, and (**K**) CD43$^+$ CD27$^+$ cells by percent of P14 and total numbers in spleen and liver at day 7. (**L**) Representative flow plots of KLRG1 and CD127 expression on P14 cells at day 31 post infection. (**M**) Representative flow plots of CD43 and CD27 expression on P14 cells at day 31 post infection. Quantification of P14 (**N**) KLRG1$^+$, (**O**) CD127$^+$ KLRG1$^-$, (**P**) CD43$^-$ CD27$^-$, and (**Q**) CD43$^+$ CD27$^+$ cells by percent of P14 and total numbers in spleen and liver at day 31 post infection. Statistics displayed determined by unpaired *t*-test between *Malat1$^{scr/scr}$* and WT transferred cells (\*p < 0.05; \*\*p < 0.01; \*\*\*p < 0.001, error bars indicate standard deviation).

these data demonstrate that miR-15/16 restrict memory cell differentiation, and reveal the ability of *Malat1*:miR-15/16 interaction to enhance memory cell differentiation across infection contexts.

### *Malat1* enhances cytotoxic T cell IL-2 production and survival

Finally, we investigated how the *Malat1*:miR-15/16 RNA circuit regulates memory cell differentiation and accumulation, giving consideration to the many direct targets of miR-15/16 and the emergent indirect effects on IL-2 and other costimulation-responsive gene expression in CD8 T cells. The pro-survival protein Bcl2 is the first characterized target of miR-15/16 (*Cimmino et al., 2005*). Higher abundance of Bcl2 in memory cells (as compared with short-lived effector cells) aids their survival and persistence, counteracting their increased expression of pro-apoptotic factors such as Bim (*Kurtulus et al., 2011*). As such, changes in the balance of pro- and anti-apoptotic factors can have selective effects on the accumulation of memory T cells. Bcl2 expression in *Malat1$^{scr/scr}$* P14 cells was reduced at day 31 in the CD43$^-$ CD27$^-$ population in both LCMV and LM-GP33 infection, and also in CD43$^+$ CD27$^+$ cells in LCMV infection (*Figure 7A–C*). This reduction is consistent with increased direct miR-15/16 action on the Bcl2 3′ UTR in the *Malat1$^{scr/scr}$* cells, but it may also be affected by IL-2 and other costimulation responsive factors. Also consistent with reduced Bcl2 expression, increased proportions of dead *Malat1$^{scr/scr}$* cells were observed in both the KLRG1$^+$ and KLRG1$^-$ cell fractions at day 7 during LM-GP33 infection (*Figure 7D, E*). Very few dead cells were detected among P14 cells of either genotype at day 31. IL-2 is critical for the induction of Bcl2 in memory T cells to promote their survival (*Toumi et al., 2022*). Furthermore, IL-2 production by cytotoxic T cells predisposes them to differentiate into a memory phenotype, likely by an autocrine/paracrine mechanism (*Kahan et al., 2022*; *Kurtulus et al., 2011*; *Wojciechowski et al., 2007*). Given the enhanced ability of miR-15/16$^{\Delta/\Delta}$ cells to produce IL-2 and the trend toward impaired IL-2 production in *Malat1$^{scr/scr}$* cells in vitro (*Figure 4D*), we assessed IL-2 production by *Malat1$^{scr/scr}$* T cells in LCMV and LM-GP33-infected mice. Corroborating our in vitro observations, a lower proportion of *Malat1$^{scr/scr}$* P14 cells produced IL-2 when stimulated ex vivo on day 7 in both infection models (*Figure 7F, G*). This defect was also shared by KLRG1$^+$ and KLRG1$^-$ populations and limited to this critical early time point. Higher proportions of memory cells were able to produce IL-2 ex vivo at day 31 after infection with either LCMV or LM-GP33 infection, and *Malat1$^{scr/scr}$* P14 cells produced equivalent amounts of IL-2 (*Figure 7H*).

As CD28 costimulation is key to IL-2 production, the observed early defect is consistent with less robust activation while antigen is still present (*Shapiro et al., 1997*). Taken together, these results indicate that *Malat1$^{scr/scr}$* cells in the context of a viral or bacterial infection receive relatively poor activating cues and subsequently produce less IL-2 early during infection, contributing to a less robust pro-survival and pro-memory state. These findings illustrate how *Malat1* regulation of miR-15/16 and its large target gene network can act through multiple connected nodes to coordinate gene expression programs essential to cytotoxic T cell responses (*Figure 8*).

## Discussion

LncRNAs are a large, diverse class of gene products that perform important physiological functions through a variety of molecular mechanisms. However, functional requirements are typically tested using RNAi to degrade the lncRNA or by disrupting their transcription entirely with no paradigm to dissect sequence-specific functions. 'Sponging' miRNAs (acting as a ceRNA) is among the most frequently proposed mechanisms of lncRNA function. Networks of non-coding ceRNAs, miRNAs, and

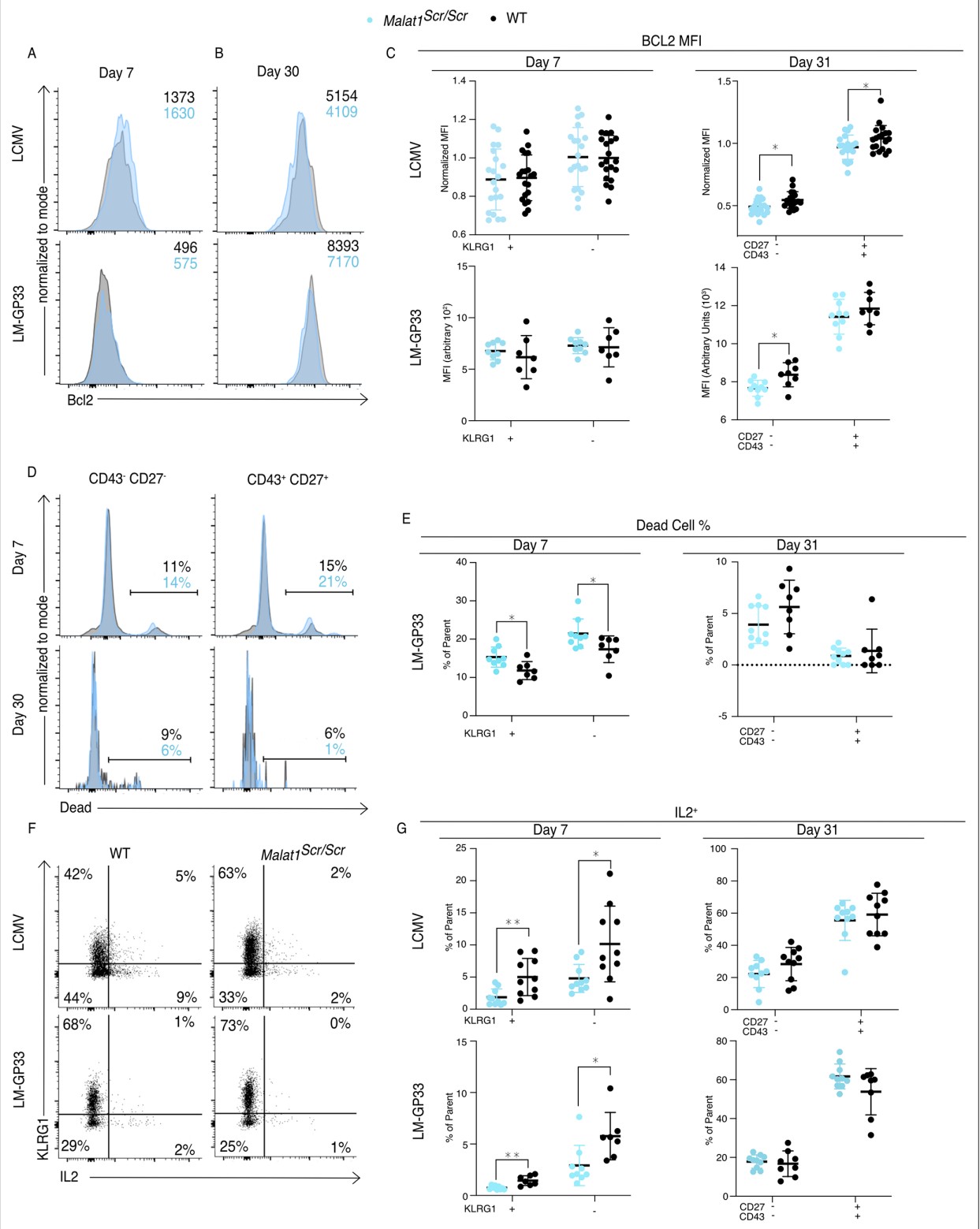

**Figure 7.** *Malat1* enhances pro-survival cues downstream of T cell activation. *Malat1scr/scr* and WT cells containing the GP33-specific TCR transgene (P14) on the CD45.2 background were transferred separately into congenic CD45.1 WT hosts. One day later the recipient mice were infected with $2 \times 10^5$ p.f.u. I.p. LCMV Armstrong or $2 \times 10^4$ c.f.u. r.o. LM-GP33. Antigen-specific responses were assayed by monitoring the transferred cells by flow cytometry in the spleen. (**A–C**) Bcl2 expression in transferred P14 cells in the spleen 7 and 31 days post infection for both LCMV and LM-GP33. Data from two independent experiments per LCMV time point and a single experiment per LM-GP33 time point. (**A**) Representative flow cytometry plots

*Figure 7 continued on next page*

*Figure 7 continued*

of P14 KLRG1+ CD127− cell Bcl2 expression 7 days post infection. Numbers shown are mean fluorescence intensity. (**B**) Representative flow cytometry plots of P14 CD43− CD27− cell Bcl2 expression 31 days post infection. Numbers shown are mean fluorescence intensity. (**C**) Quantification of Bcl2 expression producing cells by mean fluorescence intensity within the indicated P14 subpopulation defined by KLRG1 or CD27 and CD43. (**D, E**) Analysis of dead cells within splenic P14 CD43 and CD27 subpopulations at days 7 and 31 post LM-GP33 infection, data from a single experiment per time point. (**D**) Representative flow cytometry plots of P14 subsets defined by CD43 and CD27. Numbers shown represent percent of dead cells per the parent subpopulation. (**E**) Quantification of dead cells as a percentage of parent P14 subpopulation. (**F, G**) Analysis of IL-2 producing P14 subsets in the spleen via IL-2 capture assay at days 7 and 31 post infection for both LCMV and LM-GP33, data from a single experiment per infection per time point. (**F**) Representative flow cytometry plots of all P14 cells stained for KLRG1 and captured IL-2 from both infections at day 7. Numbers represent percent of cells in that quadrant of all P14 transferred cells. (**G**) Quantification of IL-2 producing cells by percent of parent within the indicated P14 subpopulation defined by KLRG1 or CD27 and CD43. Statistics displayed determined by unpaired *t*-test between *Malat1^scr/scr^* and WT transferred cells, where multiple tests were performed the Holm–Šidák method was used to correct for multiple comparisons (*p < 0.05; **p < 0.01, error bars indicate standard deviation).

target genes likely do shape gene expression programs in many biological contexts. Yet definitive evidence that a lncRNA:miRNA interaction has a physiological effect in a living organism has remained elusive. Guided by a biochemical approach, we investigated the requirements for *Malat1* interaction with miR-15/16 by surgically altering just five nucleotides within the endogenous 8 kb *Malat1* transcript in mice. Using T cells from these animals, we rigorously attribute changes in costimulation-responsive gene expression and in vivo defects in memory T cell formation to this Malat:miR-15/16 circuit.

*Malat1* is a pleiotropic lncRNA implicated in a multitude of processes, including scaffolding splicing and epigenetic regulators, binding to chromatin, and interacting with several miRNAs in different cell types (*Arun et al., 2020*). Its extremely high expression and interspecies conservation nominated Malat1 as a strong candidate to have an impact on the activity of even highly abundant miRNAs like miR-15/16 that have large effects on gene expression and cell behavior. Less abundant miRNAs can also serve essential functions (*Wigton et al., 2021*), and they are likely to be more susceptible to inhibition by *Malat1* and other ceRNAs. The same biochemically driven sequence-specific approach used here could be applied to probe the physiological effects of other lncRNA:miRNA interactions, and it could also be extended to probe the requirement for interaction with Ezh2 and other proteins.

Another important future direction is to better understand how exactly *Malat1* inhibits miR-15/16, given that TDMD does not appear to result from this interaction. Most *Malat1* resides in the nucleus, whereas miRNAs reside and function in the cytoplasm. Advances in miRNA fluorescent in situ hybridization (miR-FISH) or the implementation of proximity-CLIP (*Benhalevy et al., 2018*) may help clarify

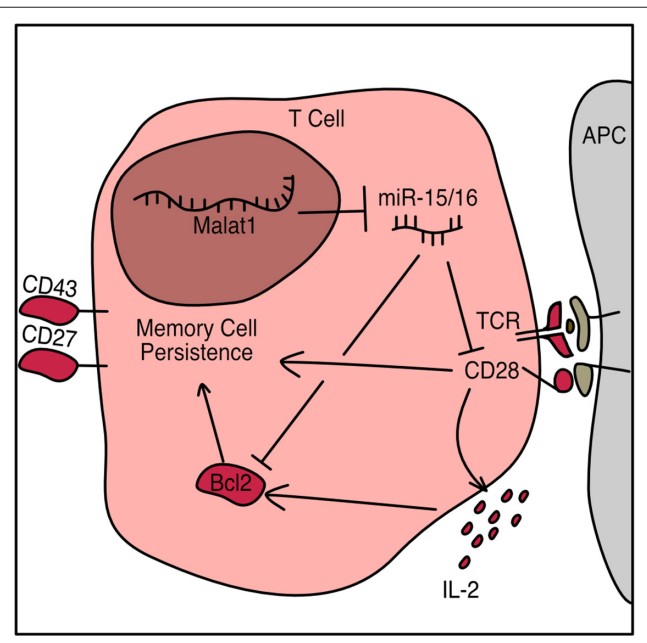

**Figure 8.** Schematic of Malat1:miR-15/16 circuit regulation of cytotoxic T cell responses.

whether *Malat1* nuclear sequestration plays a role in its ceRNA function. Detailed mechanistic understanding of this experimentally tractable circuit may illuminate how ceRNAs may be leveraged by the cell or by cellular engineers to manipulate the miRNA pool and target gene expression.

There is a growing interest in *Malat1* function within the immune system, where it is highly expressed in many cell types and often regulated in interesting ways, including differential expression in short lived effector and memory precursor cells produced by asymmetric division of activated T cells during LCMV infection (*Kakaradov et al., 2017*). Malat1 overexpression in dendritic cells promotes IL-10 production by T regulatory cells (*Wu et al., 2018*), and *Malat1*$^{-/-}$ mice exhibited reduced IL-10 and Maf expression by T helper cells in vivo and in vitro (*Hewitson et al., 2020*). Two prior reports addressed *Malat1* function in cytotoxic T cell responses. In one report, no significant differences were observed in *Malat1*$^{-/-}$ mice (*Yao et al., 2018*). It is difficult to interpret negative data generated from whole-body *Malat1* deficiency given the multitude of cell types in which *Malat1* may act. A second recent report, using RNAi to suppress Malat1 expression specifically in antigen-specific CD8$^+$ T cells, detected enhanced t-Tem differentiation (*Kanbar et al., 2022*). This observation, which contrasts with our findings using *Malat1*$^{scr/scr}$ mice, was suggested to occur via an epigenetic mechanism involving Malat1 interaction with Ezh2. This mechanism and the ceRNA function that we describe here are not mutually exclusive, and it is possible to observe divergent phenotypes from the different interventions used to probe *Malat1* function in T cells. Our *Malat1*$^{scr/scr}$ mouse combined with the cell transfer system clearly demonstrates a cytotoxic T cell-intrinsic requirement for *Malat1* to inhibit miR-15/16 to enhance activation and protect memory cell persistence. This is consistent with our previous work on the miR-15/16 family, and fitting with the increased expression of *Malat1* in memory precursors (*Gagnon and Ansel, 2019*; *Kakaradov et al., 2017*). The present study builds upon these findings and provides detailed insight into the gene regulatory programs which may drive these phenotypes. Cytotoxic T cells lacking CD28 generate similar numbers of viral specific cells at the height of infection, but fail to maintain equivalent numbers during the memory phase (*Borowski et al., 2007*). Furthermore, a major outcome of CD28 ligation is the production of IL-2, an essential T cell growth factor, and this IL-2 production is required in a cell-intrinsic manner for generation of CD127$^+$ memory cells (*Kahan et al., 2022*). Here, we demonstrate that not only does the *Malat1*-miR-15/16 circuit regulate CD28 costimulation, IL-2 production, and Bcl2 expression, but also the concordant outcome of maintaining CD43$^+$ CD27$^+$ memory cells. miR-15/16 targets a large number of genes, so it is unlikely that the effect on Bcl2 or CD28 alone is entirely responsible for the observed changes in CD43$^-$ CD27$^-$ t-TEM and CD43$^+$ CD27$^+$ memory cells. Our results are consistent with memory cells requiring a heightened level of pro-survival cues to survive the contraction bottleneck after the peak of infection. In this manner elevated *Malat1* levels can act as an enhancer of pro-survival and activating cues to prevent excessive cell death in this sensitive memory population. The magnitude of this effect was modest in acute LCMV and *Listeria* infection, two models that feature robust pathogen clearance, allowing assessment of memory T cells in the absence of chronic antigen persistence. Further work is needed to probe other settings in which *Malat1*:miR-15/16 interaction may have a bigger impact on the outcome of immune responses.

The importance of cytotoxic T cells is evident in the context of the global COVID-19 pandemic and the advent of CAR-T cell therapies reaching the clinic. Knowledge of the networks that regulate these cells and the critical nodes within these networks have the potential to augment technologies and therapies from vaccines to anti-tumor immunotherapy. Transcription factors and miRNAs have been extensively studied in this fashion as critical nodes in the regulation of gene transcription and translation. In the study presented here, we show *Malat1* acts upstream of one such node, the miR-15/16 family. From this one interaction, *Malat1* has the potential to combinatorially and synergistically regulate gene networks essential to cytotoxic T cells. This concept is easily extended when *Malat1*'s ability to regulate multiple other miRNA families is considered. If *Malat1*'s sequence-specific function is further defined, then editing or expression of specific *Malat1* sequences could be used to tune multiple miRNA families in concert while leaving other functions of this enigmatic transcript un-touched.

## Materials and methods

### Mice

WT C57BL/6 mice were bred in our facility. miR-15/16$^{\Delta/\Delta}$ and miR-15/16$^{fl/fl}$ mice were derived as described in *Gagnon and Ansel, 2019*. *Malat1$^{scr/scr}$* mice were generated from WT C57BL/6 mouse zygotes electroporated with CRISPR-Cas9 RNPs and HDR template (guide and template sequences below) as described previously (*Chen et al., 2016*). B6.SJL-Ptprca Pepcb/BoyJ (CD45.1) strain #002014 were purchased from the Jackson Laboratory. WT P14 mice carrying the TCR transgene specific to the LCMV GP33 peptide were obtained from the Waterfield lab and were backcrossed to C57BL/6 to maintain the line. *Malat1$^{scr/scr}$* mice were bred to the P14 line to generate *Malat1$^{scr/scr}$* P14 mice. Male and female age- and sex-matched mice were used between 5 and 12 weeks of age. All mice were housed and bred in specific pathogen-free conditions in the Animal Barrier Facility at the University of California, San Francisco. Animal experiments were approved by the Institutional Animal Care and Use Committee (IACUC) of the University of California, San Francisco (Protocol AN200003).

### AGO2 HITS-CLIP

CD8$^+$ T cells were isolated as below and stimulated for 3 days with αCD3 and αCD28 antibodies and grown with kool aid complete media. Subsequently, the cells were rested and expanded in kool aid complete media for 2 days supplemented with 100 U/ml recombinant Human IL-2 (R&D Systems Cat# 202-IL-010/CF). Subsequently, $1 \times 10^6$ cells were used to prepare NGS libraries as described in *Gagnon and Ansel, 2019* and *Loeb et al., 2012*. Samples were sequenced on HI-Seq 2500 (Illumina). Eleven nucleotide adaptors were trimmed from each read and resultant sequences were aligned to the mm10 genome using bowtie2 (*Langmead and Salzberg, 2012*). To assure lack of miR-15/16 binding in *Malat1$^{scr/scr}$* mice was not do to errors in alignment, reads from *Malat1$^{scr/scr}$* cells were aligned to a mm10 genome that contained a single modification changing Malat1 from the WT allele to the *Malat1$^{scr}$* allele. To determine maximum binding depth across the genome and to manipulate aligned files the samtools package was used (*Li et al., 2009*). To assess Ago2 binding, aligned HITS-CLIP reads were integrated across the follow genomic annotations: lncRNA genes from mouse Noncode v6 (http://www.noncode.org/download.php); miRNA target-binding sites from TargetScan V7.2 (https://www.targetscan.org/cgi-bin/targetscan/data_download.mmu80.cgi); mm10 introns, exons, 3' UTRs, and coding genes from UCSC genome table browser (http://genome.ucsc.edu/cgi-bin/hgGateway?db=mm10). To assess differences in target binding at specific 3' UTR sites between WT and *Malat1$^{scr/scr}$* cells, reads within each 3' UTR were normalized to the total reads contained in the given 3' UTR for each genotype. To identify regions of significant Ago2 binding above background, so-called 'peaks', we used the Piranha algorithm (*Uren et al., 2012*) (https://github.com/smithlabcode/piranha) on our HITS-CLIP. To remove sites from analysis where confounding features were present, such as miRNA expression sites or rRNA repeats, bedtools intersect was used to remove these features from annotation files (*Quinlan, 2014*). Human AHC data from publicly available datasets were trimmed and aligned to the Hg38 genome as previously described (*Karginov and Hannon, 2013*; *Li et al., 2018*).

### Cell transfer and infections

Spleens were harvested from WT or *Malat1$^{scr/scr}$* P14 CD45.2 mice into phosphate-buffered saline (PBS) and passed through 70 µm strainers to generate a single-cell suspension. Samples were then stained for live/dead (apc-cy7), Thy1.2 (BV605), CD8α (apc), and TCRVα2 (pe). Whole splenocytes were then transferred retro-orbitally (r.o.) into WT BoyJ (CD45.1) 7-week-old male recipients such that each recipient received 20,000 Thy1.2$^+$ CD8α$^+$ TCRVα2$^+$ cells in 200 µl PBS. To initiate LCMV infections, mice were injected intraperitoneally (i.p.) with $2 \times 10^5$ plaque forming units (p.f.u.) LCMV armstrong in 200 µl plain RPMI. LCMV virus was produced in house as described in *Shehata et al., 2018*. To initiate *L. monocytogenes* infection, mice were injected r.o. with $2 \times 10^5$ colony forming units (c.f.u.) *Listeria monocytogenes-GP33* in 100 µl PBS. *Listeria monocytogenes-GP33* was prepared in house as described in *Allen et al., 2020*. Blood was collected via submandibular bleeds with goldenrod 4 mm lancets collected into sample tubes coated with K2 EDTA (BD Ref# 365974) and 60 µl of blood was lysed with 500 µl of ACK lysis buffer. To assay spleens and livers, mice were sacrificed and organs harvested. To assay intracellular cytokine production, splenocytes were plated into 96-well u-bottom plates in complete kool-AID media and stimulated for 6 hr with a final concentration of 0.2 mg/ml GP33-41 (KAVYNFATM) in the presence of Brefeldin A.

## Flow cytometry

Spleens and livers were harvested into 2% fetal bovine serum (FBS) PBS with and passed through 70 μm strainers to generate a single-cell suspension. Samples were then spun at 450 r.c.f. for 5 min and livers were resuspended in 20% Percoll. Percoll suspensions were spun at 741 r.c.f. for 20 min at 25°C and the supernatants discarded. Cell pellets for both spleens and livers were resuspended in 1 ml ACK lysis buffer and incubated at 4°C for 5 min. Lysis was stopped with 5 ml 2% FBS PBS and samples again spun at 450 r.c.f. for 5 min. Samples were resuspended in 2% FBS and aliquoted into v-bottom 96 well plates. Live dead staining was then performed using the fixable viability dye at 1:2000 in PBS. Subsequently, cells were blocked with mouse Fc block 1:100 in 2% FBS PBS and stained for surface proteins with directly conjugated antibodies diluted 1:100 in 2% FBS PBS. Stains were incubated for 20 min at 4°C protected from light. For surface stains alone, LCMV samples were fixed with 4% paraformaldehyde (PFA) for 5 min at 4°C. For intracellular stains, samples were fixed and permeabilized according the the Transcription Factor FIxation Kit (Invitrogen Cat#00-5521-00). Intracellular antibodies were diluted in 1:100 in permeabilization buffer and incubated at 4°C for 30 min. All samples were spun at 821 r.c.f. for 5 min prior to being resuspended in 2% FBS PBS with 1:10 AccuCount beads (spherotech Cat #ACBP-100–10) for analysis on either the BD LSR II or the BD LSRFortessa flow cytometry analyzer.

## Cell isolation and in vitro functional assays

CD8[+] T cells were isolated from spleens using negative selection from the EasySep Mouse CD8[+] T Cell Isolation Kit (cat# 19853). Cells were counted using trypan blue staining and the nexelcom cellometer spectrum. Cultures were started via stimulation with plate-bound αCD3 (1 μg/ml, clone 2C11) and αCD28 (1 μg/ml, clone 37.51); plates coated overnight in PBS with $Ca^{2+}$ and $Mg^{2+}$ at 4°C. Cells were plated in Kool AID complete media (Dulbecco's Modified Eagle Medium (DMEM) high glucose media supplemented with 10% FBS, pyruvate, nonessential amino acids, minimum essential medium (MEM) vitamins, L-arginine, L-asparagine, L-glutamine, folic acid, beta mercaptoethanol, penicillin, and strep-tomycin) and spun at 450 r.c.f. for 5 min at 25°C to begin stimulation. For functional assays cells were harvested 2 or 4 hr after stimulation for flow cytometry, or 24 hr after stimulation for mRNA-seq. For supernatant cytokine expression, plates were spun at 450 r.c.f. for 5 min at 4°C 16 hr after stimulation and cell-free culture medium was collected and frozen at −80°C. TNFα (Cat #), IFNγ (Cat #), and IL-2 (Cat #) were analyzed by ELISA.

## miRNA qPCR

Spleens were harvested and single-cell suspensions generated by passing through 70 μm strainers. CD8+ T cells were isolated as above, and $2 \times 10^6$ cells were pelleted at 450 r.c.f. for 5 min at 4°C. Cell pellet was resuspended in 700 μl Trizol reagent (Ambion cat #15596018) and kept at –80 °C. RNA was was isolated using using the Direct-zol-96 RNA Kit (zymogen cat #R2054). This RNA was used as input to the Mir-X miRNA qRT-PCR TB Green Kit (Takara cat# 638316) to generate miRNA cDNA. Specific primers to miR-15a, miR-15b, and miR-16 were used to quantify those miRNA species on the QPCR MACHINE (eppendorf realplex²). ribosomalRNA 5.8 s was used as a housekeeping control for each sample. Each sample was run in duplicate. To quantify miRNA expression technical duplicates were averaged and then normalized to rRNA 5.8 s by subtracting the rRNA 5.8 s Ct value from miRNA Ct value (ΔCt). Expression values reported are generated by $2^{-\Delta Ct}$.

## mRNA sequencing

$1 \times 10^6$ CD8[+] T cells were harvested 24 hr after αCD3 ± αCD28 stimulation as described above. Cells were pelleted at 450 r.c.f. for 5 min at 4°C. Cell pellet was resuspended in 700 μl Trizol reagent (ambion ref #15596018) and kept at −80°C. RNA was was isolated using using the Direct-zol-96 RNA Kit (zymogen cat #R2054). The integrity of total RNA was checked on Fragment Analyzer (Agilent, Cat. No. DNF-472), only RNA with RQN number of above 7 was used for library construction. The starting quantity of 100 ng of total RNA was used according to vendor instructions with Universal plus mRNA with Nu Quant (TECAN, Cat. No. 0520), final library PCR amplification was 17 cycles. After library completion, individual libraries were pooled equally by volume, and quantified on Fragment Analyzer (Agilent, Cat. No. DNF-474). Quantified library pool was diluted to 1 nM and further diluted as per protocol and sequenced on Illumina MiniSeq (Illumina, Cat. No. FC-420-1001) to check for quality

of reads. Finally, individual libraries were normalized according to MiniSeq output reads, specifically by % protein-coding genes and were sequenced on one lane of NovaSeq6000 S4 PE100 (Illumina, Cat. No. 20028313). Reads were aligned to the mouse genome GRCm38 and quantified using the STAR aligner software version 2.7.2b. Read normalization and differential expression analysis were performed in the R computing environment version 3.6.1 using the software DESeq2 version 1.26. For RNA sequencing analysis, FDR-corrected p-values were used to evaluate significant differences between experimental groups using a significance threshold of 0.05. Lowly expressed genes that had fewer than 2 reads per million on average across all samples were removed from the analysis. Empirical cumulative density plots were made using the ggplot package for R. Heatmaps and hierarchical clustering were performed via the gplots package for R. Gene ontology analysis was performed using the statistical overrepresentation test for Panther pathways (*Mi et al., 2019*).

## Materials availability
*Malat1^scr/scr^* mice will be provided upon request.

## Statistical and analytical software
All flow cytometry data were analyzed using FlowJo (version 10). Statistical analyses and plotting were performed using GraphPad Prism (Version 9.2.0) and R (version 4.2.1).

## Data availability and software
Ago2 HITS-CLIP data uploaded to NCBI GEO accession #GSE216565.

mRNA sequencing data uploaded to NCBI GEO accession #GSE216113.

Code used to analyze HITS-CLIP data summed over annotations and peaks in WT cells can be found at https://github.com/AnselLab/WT_HITS_CLIP_Analyses, (copy archived at *Wheeler, 2023a*).

Code used to analyze relative binding densities of HITS-CLIP reads can be found at https://github.com/AnselLab/Malat1_miR1516_AGO2_HITS_CLIP, (copy archived at *Wheeler, 2023b*).

Code used to analyze mRNAseq data for both miR-15/16 target expression and CD28 responsive genes can be found at https://github.com/AnselLab/Malat1_miR1516_CD3CD28_RNAseq, (copy archived at *Wheeler, 2023c*).

## Acknowledgements
We thank Shomyseh Sanjabi, Nadia Roan, and Jason Neidleman for generous donation of the LCMV armstrong virus; Michael Waterfield for generous donation of P14 mice; Marlys Fasset, Simon Zhou, and Eric Wigton for maintaining mutant mouse colonies through the COVID-19 pandemic; Walter Eckelbar, Eunice Wan, Charina Julian, Lenka Maliskova, and Andrew Schroeder for assistance with mRNA-seq and the sequencing of our HITS-CLIP libraries. This work was supported by the NIH (HL109102 and HL107202), the Sandler Asthma Basic Research Center, and the Hooper Foundation. Priscila Muñoz-Sandoval is a Howard Hughes Medical Institute Gilliam Fellow. We acknowledge the PFCC (RRID:SCR_018206) for assistance generating flow cytometry data. Research reported here was supported in part by the DRC Center Grant NIH P30 DK063720.

## Additional information

### Competing interests
John D Gagnon: JDG is now an employee of Arsenal Biosciences. Matthew H Spitzer: M.H.S. is founder and a board member of Teiko.bio and has received a speaking honorarium from Fluidigm Inc, has served as a consultant for Five Prime, Ono, January, Earli, Astellas, and Indaptus, and has received research funding from Roche/Genentech, Bristol Myers Squibb, Valitor, and Pfizer. Alexander Marson: A.M. is a cofounder of Arsenal Biosciences, Spotlight Therapeutics, and Survey Genomics, serves on the boards of directors at Spotlight Therapeutics and Survey Genomics, is a board observer (and former member of the board of directors) at Arsenal Biosciences, is a member of the scientific advisory boards of Arsenal Biosciences, Spotlight Therapeutics, Survey Genomics, NewLimit, Amgen and Tenaya, owns stock in Arsenal Biosciences, Spotlight Therapeutics, NewLimit, Survey Genomics,

PACT Pharma, and Tenaya and has received fees from Arsenal Biosciences, Spotlight Therapeutics, NewLimit, 23andMe, PACT Pharma, Juno Therapeutics, Trizell, Vertex, Merck, Amgen, Genentech, AlphaSights, Rupert Case Management, Bernstein and ALDA. A.M. is an investor in and informal advisor to Offline Ventures and a client of EPIQ. The Marson laboratory has received research support from Juno Therapeutics, Epinomics, Sanofi, GlaxoSmithKline, Gilead and Anthem. The other authors declare that no competing interests exist.

## Funding

| Funder | Grant reference number | Author |
| --- | --- | --- |
| National Heart, Lung, and Blood Institute | R01 HL109102 | K Mark Ansel |
| National Heart, Lung, and Blood Institute | P01 HL107202 | K Mark Ansel |
| National Institute of Diabetes and Digestive and Kidney Diseases | P30 DK063720 | Matthew H Spitzer Alexander Marson K Mark Ansel |
| Sandler Foundation | Sandler Asthma Basic Research Center investigator support | K Mark Ansel |
| Howard Hughes Medical Institute | Gilliam Fellowship | Priscila Muñoz-Sandoval K Mark Ansel |
| Hooper Foundation | Graduate student fellowships | Benjamin D Wheeler Wandi S Zhu |

The funders had no role in study design, data collection, and interpretation, or the decision to submit the work for publication.

## Author contributions

Benjamin D Wheeler, Conceptualization, Data curation, Software, Formal analysis, Validation, Investigation, Visualization, Methodology, Writing - original draft, Writing – review and editing; John D Gagnon, Conceptualization, Data curation, Software, Formal analysis, Investigation, Visualization, Methodology, Writing – review and editing; Wandi S Zhu, Priscila Muñoz-Sandoval, Data curation, Investigation, Writing – review and editing; Simon K Wong, Rachel DeBarge, Data curation, Writing – review and editing; Dimitre S Simeonov, Conceptualization, Data curation, Investigation, Methodology, Writing – review and editing; Zhongmei Li, Data curation, Methodology, Writing – review and editing; Matthew H Spitzer, Alexander Marson, Resources, Supervision, Funding acquisition, Methodology, Writing – review and editing; K Mark Ansel, Conceptualization, Resources, Formal analysis, Supervision, Funding acquisition, Investigation, Visualization, Methodology, Writing - original draft, Project administration, Writing – review and editing

## Author ORCIDs

Benjamin D Wheeler http://orcid.org/0000-0001-5310-7213
John D Gagnon http://orcid.org/0000-0001-6208-5781
Rachel DeBarge http://orcid.org/0000-0001-9223-1364
K Mark Ansel http://orcid.org/0000-0003-4840-9879

## Ethics

All mice were housed and bred in specific pathogen-free conditions in the Animal Barrier Facility at the University of California, San Francisco. Animal experiments were approved by the Institutional Animal Care and Use Committee (IACUC) of the University of California, San Francisco, protocol number AN2000003.

Reviewer #1 (Public Review): https://doi.org/10.7554/eLife.87900.3.sa1
Reviewer #2 (Public Review): https://doi.org/10.7554/eLife.87900.3.sa2
Author Response https://doi.org/10.7554/eLife.87900.3.sa3

## Additional files

### Supplementary files
• MDAR checklist

### Data availability
Ago2 HITS-CLIP data deposited in GEO under accession code #GSE216565. mRNA sequencing data deposited in GEO under accession code #GSE216113.

The following datasets were generated:

| Author(s) | Year | Dataset title | Dataset URL | Database and Identifier |
|---|---|---|---|---|
| Wheeler BD, Ansel KM, Gagnon JD, Zhu WS, Muñoz-Sandoval P, Simeonov D, Li Z, Marson A, Spitzer M, Wong SK, Debarge R | 2022 | The lncRNA Malat1 Inhibits miR-15/16 to Enhance Cytotoxic T Cell Activation and Memory Cell Formation [Ago2 HITS-CLIP] | https://www.ncbi.nlm.nih.gov/geo/query/acc.cgi?acc=GSE216565 | NCBI Gene Expression Omnibus, GSE216565 |
| Wheeler BD, Ansel KM, Gagnon JD, Zhu WS, Muñoz-Sandoval P, Simeonov D, Li Z, Marson A, Spitzer M, Wong SK, Debarge R | 2022 | The lncRNA Malat1 Inhibits miR-15/16 to Enhance Cytotoxic T Cell Activation and Memory Cell Formation | https://www.ncbi.nlm.nih.gov/geo/query/acc.cgi?acc=GSE216113 | NCBI Gene Expression Omnibus, GSE216113 |

The following previously published datasets were used:

| Author(s) | Year | Dataset title | Dataset URL | Database and Identifier |
|---|---|---|---|---|
| Karginov FV, Hannon GJ | 2013 | Remodeling of Ago2-mRNA interactions upon cellular stress reflects miRNA complementarity and correlates with altered translation rates | https://www.ncbi.nlm.nih.gov/geo/query/acc.cgi?acc=GSE44404 | NCBI Gene Expression Omnibus, GSE44404 |
| Li Y, Estep JA, Karginov FV | 2018 | Transcriptome-wide identification and validation of interactions between the miRNA machinery and HuR on mRNA targets | https://www.ncbi.nlm.nih.gov/geo/query/acc.cgi?acc=GSE102321 | NCBI Gene Expression Omnibus, GSE102321 |

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

# Appendix 1

## Appendix 1—key resources table

| Reagent type (species) or resource | Designation | Source or reference | Identifiers | Additional information |
|---|---|---|---|---|
| Commercial assay or kit | IL-2 ELISA | Invitrogen | 5018280 | |
| Commercial assay or kit | TNFα ELISA | Invitrogen | BMS607-3 | |
| Commercial assay or kit | IFNγ ELISA | R&D Systems | MIF00 | |
| Commercial assay or kit | IL-2 Secretion Assay APC | Miltenyi Biotec | 130-090-987 | |
| Sequence-based reagent | Malat1 CRISPR Guide 1 | Dharmacon | RNA oligomer | Sequence – GCATTCTAATAGCAGCAGAT |
| Sequence-based reagent | Malat1 HDRT Ultramer | IDT | DNA oligomer | Sequence – ACAGACCACACAGAATGCAGGTGTCTTG ACTTCAGGTCATGTCTGTTCTTTGGCAAGTAATATGT GCAGTACTGTTCCAATCTGTCCTGATTAGAATGCATT GTGACGCGACTGGAGTATGATTAAAGAAAGTTGTG TTTCCCCAAGTGTTTGGAGTAGTGGTTGTTGGAGG AAAAGCCATGAGTAACAGGCTGAGTGTT |
| Antibody | anti-CD127 PE (Clone-A7R34, rat monoclonal) | Invitrogen | 12-1271-83 | (1:200) |
| Antibody | anti-CD127 FITC (Clone-A7R34, rat monoclonal) | Invitrogen | 11-1271-82 | (1:200) |
| Antibody | anti-CD45.2 BV785 (Clone-104, mouse monoclonal) | BioLegend | 109839 | (1:200) |
| Antibody | anti- CD45.1 Alexa Fluor 700 (Clone-A20, mouse monoclonal) | BioLegend | 110724 | (1:200) |
| Antibody | anti-CD90.2 BV605 (Clone-30-H12, rat monoclonal) | BD Biosciences | 740334 | (1:200) |
| Antibody | anti-CD90.2 eFluor 450 (Clone-53-2.1, rat monoclonal) | eBiosciences | 48-0902-80 | (1:200) |
| Antibody | anti-CD44 PE-Cy7 (Clone-IM7, rat monoclonal) | eBiosciences | 25-0441-82 | (1:200) |
| Antibody | anti-CD27 APC (Clone-LG.7F9, Armenian hamster monoclonal) | eBiosciences | 17-0271-82 | (1:200) |
| Antibody | anti-CD27 FITC (Clone-LG.7F9, Armenian hamster monoclonal) | eBiosciences | 11-0271-82 | (1:200) |
| Antibody | anti-Bim PE (Clone-C34C5, rabbit monoclonal) | Cell Signaling | 12186S | (1:100) |
| Antibody | anti-CD43 Percp-Cy5.5 (Clone-1B11, rat monoclonal) | BioLegend | 121224 | (1:200) |
| Antibody | anti-CD45.2 PE-Cy7 (Clone-104, Mouse monoclonal) | BD Biosciences | 560696 | (1:200) |
| Antibody | anti-CD45.1 BV785 (Clone-A20, mouse monoclonal) | BioLegend | 110743 | (1:200) |

*Appendix 1 Continued on next page*

*Appendix 1 Continued*

| Reagent type (species) or resource | Designation | Source or reference | Identifiers | Additional information |
|---|---|---|---|---|
| Antibody | anti-Bcl2 Alexa Fluor 647 (Clone-BCL/10C4) | BioLegend | 633510 | (1:100) |
| Antibody | anti-Bim Alexa Fluor 700 (Clone-C34C5, rabbit monoclonal) | Cell Signaling | 28997S | (1:100) |
| Other | GP33 Tetramer PE | NIH Tetramer Core | 57624 | Peptide: KAVYNFATM (1:100) |
| Antibody | anti-KLRG1 BV711 (Clone-2F1, hamster monoclonal) | BD Bioscience | 564014 | (1:200) |
| Other | Fixable Viability Dye eFluor 780 | eBioscience | 65-0865-14 | (1:2000) |
| Antibody | anti-CD8α BV805 (Clone-53-6.7, rat monoclonal) | BD Bioscience | 612898 | (1:200) |
| Antibody | anti-CD28 FITC (Clone-E18, mouse monoclonal) | BioLegend | 122008 | (1:200) |
| Antibody | anti-CD69 APC (Clone-H1.2F3, Armenian hamster monoclonal) | Invitrogen | 17-0691-82 | (1:200) |
| Antibody | anti-Nur77 PE (Clone-12.14, mouse monoclonal) | Invitrogen | 12-5965-80 | (1:100) |
| Antibody | anti-CD62L BV605 (Clone-MEL-14, rat monoclonal) | BD Biosciences | 563252 | (1:200) |

