## [Editor Report · eLife assessment]

This is an **important** study that revealed a new noncoding RNA regulatory circuit involved in T cell function. The authors provide **compelling** evidence, that is more rigorous than the state-of-the-art, using genetically engineered mice and cell-based experiments. The interpretation of the results should be tempered due to the small effect size observed.

---

## [Referee Report · Reviewer #1 (Public Review)]

Wheeler et al. have discovered a new RNA circuit that regulates T-cell function. They found that the long non-coding RNA Malat1 sponges miR-15/16, which controls many genes related to T cell activation, survival, and memory. This suggests that Malat1 indirectly regulates T-cell function. They used CRISPR to mutate the miR-15/16 binding site in Malat1 and observed that this disrupted the RNA circuit and impaired cytotoxic T-cell responses. While this study presents a novel molecular mechanism of T-cell regulation by Malat1-miR-15/16, the effects of Malat1 are weaker compared to miR-15/16. This could be due to several reasons, including higher levels of miR-15/16 compared to Malat1 or Malat1 expression being mostly restricted to the nucleus. Although the role of miR15/16 in T-cell activation has been previously published, if the authors can demonstrate that miR15/16 and/or Malat1 affect the clearance of Listeria or LCMV, this will significantly add to the current findings and provide physiological context to the study.

---

## [Referee Report · Reviewer #2 (Public Review)]

This study connects prior findings on MicroRNA15/16 and Malat1 to demonstrate a functional interaction that is consequential for T cell activation and cell fate.

The study uses mice (Malat1scr/scr) with a precise genetic modification of Malat1 to specifically excise the sites of interaction with the microRNA, but sparing all other sequences, and mice with T-cell specific deletion of miR-15/16. The effects of genetic modification on in vivo T-cell responses are detected using specific mutations and shown to be T-cell intrinsic.

It is not known where in the cell the consequential interactions between MicroRNA15/16 and Malat1 take place. The authors depict in the graphical abstract Malat1 to be a nuclear lncRNA. Malat 1 is very abundant, but it is unclear if it can shuttle between the nucleus and cytoplasm. As the authors discuss future work defining where in the cell the relevant interactions take place will be important.

In addition to showing physiological phenotypic effects, the mouse models prove to be very helpful when the effects measured are small and sometimes hard to quantitate in the context of considerable variation between biological replicates (for example the results in Figure 4D).

The impact of the genetic modification on the CD28-IL2- Bcl2 axis is quantitatively small at the level of expression of individual proteins and there are likely to be additional components to this circuitry.

---

## [Author Response]

**Review 1:**
Major concerns that need to be addressed:Investigate the effects of Malat1 on the clearance of Listeria or LCMV.

In our prior publication (Gagnon et al, Cell Reports) we showed that miR-15/16 deficiency in T cells does not affect the clearance of LCMV, and that transferred memory T cells formed in these mice can function normally to clear a secondary infection with Listeria expressing the LCMV gp33 peptide. However, the size of the memory pool was clearly changed, as was the programming of memory cells. Here, we show that disrupting miR15/16 binding to MALAT1 induces a reciprocal phenotype, validating a biological function for this RNA:RNA interaction. We employed these systems because they are widely used to reveal key aspects of T cell memory, but both infections are readily cleared by the host. These changes in the memory response likely play a limiting role in some biological context(s), and we agree that further investigation to uncover such situations would further validate the importance of this RNA circuit.

Demonstrate that Malat1 shuttles to the cytosol, this will strengthen the conclusions that Malat1 sponges miR15/16.

The location of miR-15/16 interaction with Malat1 is an interesting area for future study. Many prior studies have shown clearly that Malat1 is primarily located in the nucleus, but since T cells express such a large excess of this lncRNA, even the remaining fraction detected in the cytosol may be sufficient to “sponge” a significant amount of miR-15/16. Alternatively, these molecules may interact in the nucleus, or during mitosis. As the reviewer suggests, Malat1 may shuttle between compartments, raising the intriguing possibility that it could not only “sponge” but “drag” miR-15/16 away from its targets into the nucleus. A proper analysis of the mechanism of ceRNA function is beyond the scope of this paper, but we do believe that this circuit may be an especially good one for further study.

Through flow cytometry or immunoblot analyses, investigate the effects of Malat1-miR15/16 on genes listed in table 3. This would add credence to the sequencing and CLIP data.

We thank the reviewer for bringing to our attention the manuscript’s overemphasis on the former Table 3 gene set, which represented just a few of the hundreds of genes for which our data provide evidence for miR-15/16 binding and inhibition of expression. We have removed this table to avoid the appearance of suggesting an oversimplified model for how miR-15/16 regulate T cell responses, and replaced it with a short description of two targets (Pik3r1 and Mapk8) that link the roles of miR-15/16 in T cell activation and tumor suppression. Like transcription factors, miRNAs function as network regulators of gene expression, gaining biological power through their ability to coregulate many genes with convergent effects on cell behavior. In the case of miR-15/16, our published data, reinforced by the data in this manuscript, indicates that the relevant target network is very large, and that even very small changes in the expression of these targets is sufficient to alter the fate of antigen-responsive T cells in the setting of acute infection.

This comment also raises the important issue of target validation, which is often difficult, since the effect size for each miRNA target is small (typically 10-30%, sometimes reaching 50% reduction). The expected effect of Malat1 inhibition of miR-15/16 is some fraction of that. Nevertheless, in Figure 3 and Figure 7, we validated two direct targets (CD28 and Bcl2) using flow cytometry, a technique that facilitates precise sampling of protein expression on a large number of individual cells.

Minor concerns:The discussion is too broad and does not address the limitations of the study.

We added a sentence to acknowledge the limitation regarding small effect sizes and the shortcomings of the acute infection models used in this study:

“The magnitude of this effect was modest in acute LCMV and Listeria infection, two models that feature robust pathogen clearance, allowing assessment of memory T cells in the absence of chronic antigen persistence. Further work is needed to assess other settings in which Malat1:miR-15/16 interaction may have a bigger impact on the outcome of immune responses.”

**Reviewer 2:**
1. Given the lack of an effect on microRNA or Malat1 levels following the genetic modification is it possible that Malat1 is actually not directly bound by the miRNA? Could the knock-out of the miRNA could induce Ago2 loss on Malat1 by indirect mechanisms? If there is any room for doubt about a direct interaction the authors should at least mention discuss.

There is very little room for doubt about the direct interaction between miR-15/16 and Malat1. The AHC data we report indicates that the loss of Ago2 binding to the mutant Malat1 occurs predominantly at the site containing the miR-15/16 binding site of interest. This suggests that the mutation we created does not affect global Ago2 levels or occupancy across the rest of the transcript. Further, the miR-15/16 KO data directly support this result, showing that miR-15/16 is necessary for Ago2 binding at that site. If loss of miR15/16 resulted in a non-specific indirect loss of binding to Malat1, we would expect that other binding events would be affected as well, which we do not observe.

In the Results, the authors write: "miR-15/16 has not been previously shown to interact with Malat1", but they should cite/discuss: MALAT1 regulates the transcriptional and translational levels of proto-oncogene RUNX2 in colorectal cancer metastasis, Qing Ji et al, 2019.

We thank the reviewer for bringing this study to our attention, and we have cited it in our updated version of the manuscript. While the interaction between miR-15/16 and Malat1 has been shown before, our study represents a significant step beyond this study in two important ways: The rigorous biochemical mapping of the miR-15/16:Malat1 interaction site, and direct evidence for the role of a miR:lncRNA interaction in an in vivo physiological phenotype.

2. The authors write: "Only a few studies demonstrate sequence dependent function of lncRNAs (Elguindy and Mendell, 2021; Kleaveland et al., 2018; Lee et al., 1999)". But this seems more common that the statement implies (see for example this review: https://www.sciencedirect.com/science/article/pii/S002228361200896 0#s0065).Moreover, SNPs in lncRNAs are associated with pathologies (see for example:https://www.ncbi.nlm.nih.gov/pmc/articles/PMC6306726/, where also SNPs in Malat1 are presented). The authors could acknowledge this and by reformulating their sentence and citing these.

A large number of studies uncovered lncRNA functions without identifying RNA sequences that are responsible for that activity, but evidence for sequence-specific effects remain rare. We thank the reviewer for providing direction to additional sequence-specific studies and we have now cited several of them in the updated version of the introduction:

“Studies demonstrating sequence dependent function of lncRNAs are comparatively rare(Carrieri et al., 2012; Elguindy and Mendell, 2021; Faghihi et al., 2008; Gong and Maquat, 2011;Kleaveland et al., 2018; Lee et al., 1999; Yoon et al., 2012).”

In particular, association of important SNPs with lncRNA loci is an exciting motivator in the study of lncRNAs and can be informative in the dissection of lncRNA function. For Malat1 in the linked Minotti et al publication, we do not believe the SNPs referenced represent indications of sequence-specific transcript function. The SNPs identified for Malat1 are rs1194338, rs4102217, and rs591291. In the UCSC genome browser screenshot in Author response image 1, you can see that all of these SNPs are upstream of Malat1 and in regions of extremely dense H3K27Ac, suggesting enhancer function. These SNPs do not represent sequence specific function of the Malat1 transcript, but rather more likely genomic sequence regulation of Malat1 (or nearby gene) expression.

3. Figure 2H: In the figure legend, could the authors clarify what they mean by "same conditions as in F"?

We have updated the figure legend for clarity.

4. Figure 3 panel labels B, C, D don't match figure.

We have corrected this and provided an updated figure.

5. Figure 4 D, E, F: Can the authors comment more about why in their opinion early activation genes are not significantly decreased in Malat1 scr/scr?

Figure 4A shows that interrupting Malat1 interaction with miR-15/16 does affect the early induction of the immediate early gene CD69. Even miR-15/16 deficiency did not affect Nur77 expression, indicating that Malat1 and miR-15/16 regulate specific cues and signaling pathways involved in T cell activation. In particular, the transcriptomic analysis led us to focus on effects on costimulation-induced genes (Figure 3). Figure panels 4D, E, and F show the production of cytokines, including IL-2, which has been well documented to be responsive to CD28 signaling and clearly did so in our experiments. These data show a consistent increase in miR-15/16-deficient T cells, despite considerable noise in the assay. The trend toward reduced IL-2 in Malatscr/scr T cells is of smaller magnitude, as expected, and not statistically significant. Repeating this assay to obtain a better p value doesn’t seem warranted. However, we did independently observe decreased IL-2 production in Malatscr/scr T cells in an ex vivo cytokine capture assay (Figure 7F-G).